# RFD-LoRA: Robust Federated Distillation for LoRA Fine-Tuning under Heterogeneous and Adversarial Clients

## Abstract

Federated learning (FL) with low-rank adaptation (LoRA) is attractive for efficiency but fragile compared to full-rank FL. We show three fundamental vulnerabilities: (i) aggregation and projection bias, since bilinear averaging of adapters misrepresents the true global update; (ii) adversarial amplification, where low-rank projections can magnify malicious perturbations; and (iii) Jacobian sensitivity, where small adapter changes trigger large gradient variation. Existing methods only mitigate these issues and require identical client ranks, limiting practicality. We propose Robust Federated Distillation for LoRA (RFD-LoRA), the first framework to combine federated distillation with LoRA. By aggregating logits in a shared subspace, RFD-LoRA totally eliminates aggregation and initialization lag while enabling clients with heterogeneous ranks and adapter structures to collaborate seamlessly. To defend against non-IID and adversarial clients, we design three modules: Confidence-Adaptive Temperature (CAT), MMD-based Distillation (MMD-KD), and Disagreement Suppression (DIS). We provide error bounds and show on GLUE benchmarks that RFD-LoRA consistently outperforms prior methods in accuracy and robustness.

## 1 Introduction

Federated learning (FL) (Konečný et al. (2016)) has become a central paradigm for collaborative training across distributed data silos while preserving data privacy (McMahan et al. (2023); Kairouz et al. (2021); Ye et al. (2024)). In parallel, low-rank adaptation (LoRA) has emerged as a leading approach for parameter-efficient fine-tuning (PEFT) (Han et al. (2024); Houlsby et al. (2019)) of large pre-trained models (Hu et al. (2021); Asadi et al. (2024)), significantly reducing both computational and storage costs. Recently, these two directions have begun to converge, giving rise to the study of federated LoRA fine-tuning, where only the lightweight low-rank adapters are exchanged across clients rather than full model parameters. A number of methods have been proposed in this space, including FFA-LoRA (Sun et al. (2024)), which introduces privacy-preserving and fairness-adaptive aggregation for federated LoRA, LoRA-FAIR (Bian et al. (2025)), which improves fairness and communication efficiency across heterogeneous clients, and FLoRA (Wang et al. (2024)), which establishes the first baseline framework for federated LoRA fine-tuning. More recent works include FedIT (Zhang et al. (2024)), which addresses initialization mismatch and aggregation bias in LoRA modules, and FlexLoRA (Bian et al. (2025)), which enables flexible low-rank adaptation across heterogeneous client devices. Together, these studies demonstrate the potential of combining FL and LoRA to achieve efficient, scalable, and privacy-preserving model adaptation.

Despite these advances, federated LoRA remains fundamentally fragile. Current methods inherit three key issues: **(i) Aggregation bias**: as server-side averaging of low-rank matrices $\bar{A}, \bar{B}$ does not equal the true weighted average $\sum_k p_k B_k A_k$ (Sun et al. (2024)), and **(ii) Initialization bias**: as methods like FLoRA reinitialize adapters every round, leading to poor conditioning and delayed convergence (Bian et al. (2025)). **(iii) Identical client model structure**: existing frameworks assume that all clients use identical LoRA rank and adapter structure. This assumption may hold in simulation benchmarks, but is unrealistic in real-world federated environments where clients have diverse hardware constraints and heterogeneous adaptation needs. As a result, current LoRA-FL solutions face severe scalability and deployment challenges.

More critically, FL LoRA fine-tuning amplifies sensitivity to heterogeneity (Wang et al. (2022); Zhao et al. (2018); Li et al. (2020); Bhagoji et al. (2019)) in ways that are not present in full-rank FL. Because adapters project updates into low-dimensional subspaces, client differences are not smoothed out but instead magnified. We identify three fundamental sources of vulnerability:

**(i) Projection bias.** Server-side aggregation of LoRA modules is inherently inconsistent: $\sum_k p_k B_k A_k \neq (\sum_k p_k B_k)(\sum_k p_k A_k)$. This bilinear mismatch leads to systematic deviation between the aggregated update and the true client-average update direction.

**(ii) Adversarial amplification.** Malicious clients (Chacko et al. (2024); Tsipras et al. (2019)) align updates with adapter subspaces, yielding up to $\Theta(\sqrt{d/r})$ magnification over full-rank.[1]

**(iii) Jacobian sensitivity.** The bilinear map $(A, B) \mapsto BA$ induces Jacobian norms that scale with the feature energy $\|x\|$ (Novak et al. (2018); Moosavi-Dezfooli et al. (2018)), the adapter spectral norms, and downstream layer norms. Consequently, even small perturbations in $(A, B)$ can cause disproportionately large variations in gradients, making federated LoRA especially unstable under non-IID or adversarial settings.

In this work, we propose RFD-LoRA, the first framework that integrates federated distillation Lin et al. (2021); Itahara et al. (2023) with LoRA fine-tuning. Unlike prior methods that aggregate adapter parameters, RFD-LoRA operates in logit space: each client transmits logits on a small public anchor set, which are aggregated at the server. This eliminates aggregation bias and initialization lag from low-rank mismatches, and enables heterogeneous clients with different ranks or adapter structures to participate seamlessly via a shared latent logit subspace. Moreover, to strengthen robustness, we introduce three modules: **(i) Confidence-Adaptive Temperature (CAT)** dynamically scales logits by confidence, bounding gradient norms and stabilizing optimization; **(ii) MMD-based Distillation (MMD-KD)** aligns both mean and variance of logits, resisting energy-shaping attacks; **(iii) Disagreement Suppression (DIS)** downweights clients with high variance on anchor predictions, mitigating non-IID amplification. Together, these components yield provable error bounds and robustness against adversarial and heterogeneous clients. By uniting logit-space distillation with these defenses, RFD-LoRA achieves communication efficiency while, for the first time, enabling rank-flexible and robust federated LoRA training.

Our key contributions are summarized as follows:

- We introduce the first federated distillation framework for LoRA, which removes aggregation bias and initialization lag by aggregating logits instead of adapter parameters.
- RFD-LoRA enables heterogeneous LoRA ranks and adapter structures through a shared latent logit subspace, making it practical for real-world federated settings with diverse client resources.
- We present the first theoretical analysis of federated LoRA fragility, characterizing projection bias, adversarial amplification, and Jacobian sensitivity, and proving error bounds that quantify their impact.
- To improve robustness, we design three modules—Confidence-Adaptive Temperature (CAT), MMD-based Distillation (MMD-KD), and Disagreement Suppression (DIS)—and show through GLUE experiments that RFD-LoRA consistently outperforms existing baselines under IID/non-IID and adversarial clients.

## 2 PRELIMINARIES

We formalize federated learning with LoRA fine-tuning and show why it is inherently fragile for parameter-space training, especially under heterogeneous or adversarial clients.

### 2.1 AGGREGATION BIAS AND INITIALIZATION LAG

A central difficulty in federated LoRA fine-tuning lies in the gap between the server-reconstructed update and the true global update. In frameworks such as FedIT (Zhang et al. (2024)), each client $k$

---

[1]$d$ is the layer dimension; $r \ll d$ is the LoRA rank in $W = W_0 + \frac{\alpha}{r} BA$, $A \in \mathbb{R}^{r \times d}$, $B \in \mathbb{R}^{d \times r}$. Intuition: energy concentrates from $d$ to $r$ dimensions, giving a $\sqrt{d/r}$ ratio.

adapts a shared pre-trained model $W$ by learning local low-rank factors $A_k \in \mathbb{R}^{r \times d}$ and $B_k \in \mathbb{R}^{d \times r}$ on its private dataset $\mathcal{D}_k$. After training, the server performs data-size weighted averaging,

$$\bar{A} = \sum_{k=1}^{K} p_k A_k, \qquad \bar{B} = \sum_{k=1}^{K} p_k B_k, \qquad p_k = \frac{|\mathcal{D}_k|}{\sum_{j=1}^{K} |\mathcal{D}_j|},$$

and then forms the aggregated update as

$$\Delta W' = \bar{B}\bar{A} = \Big( \sum_{k=1}^{K} p_k B_k \Big) \Big( \sum_{k=1}^{K} p_k A_k \Big). \tag{1}$$

The ideal update, however, should be

$$\Delta W = \sum_{k=1}^{K} p_k B_k A_k, \tag{2}$$

which generally differs from $\Delta W'$ because matrix multiplication is bilinear rather than linear. We refer to $\Delta W' - \Delta W$ as the aggregation bias, which grows with diversity in client-specific adapters and is particularly problematic under non-IID data or heterogeneous ranks. Another major difficulty in federated LoRA fine-tuning comes from the way clients initialize their LoRA modules at the start of each training round, which leads to initialization lag. See details in Appendix A.

## 2.2 PROJECTION BIAS FROM SUBSPACE MISALIGNMENT

In LoRA fine-tuning, each client update is confined to a rank-$r$ subspace defined by its adapters. Let the full gradient be $g_k = \nabla f_k(W) \in \mathbb{R}^{d \times d}$. Since only $(A_k, B_k)$ are trainable, the effective update is a two-sided projection:

$$\Delta W_k \approx -\eta\, P_k(g_k), \quad P_k(X) = P_{U_k} X P_{V_k},$$

where $P_{U_k}$ projects onto $\mathrm{span}(B_k)$ and $P_{V_k}$ onto $\mathrm{span}(A_k^\top)$. The server aggregates as

$$\Delta W_{\mathrm{agg}} = -\eta \cdot \frac{1}{K} \sum_{k=1}^{K} P_k(g_k),$$

while the ideal full-model update is $\Delta W^\star = -\eta \nabla F(W)$. Thus, we define the projection bias

$$\mathcal{B}_{\mathrm{proj}} := \frac{1}{K} \sum_{k=1}^{K} P_k(g_k) - \nabla F(W).$$

If all clients share the same subspace $P$, then $\frac{1}{K}\sum_k P_k(g_k) = P(\nabla F(W))$, with error only from truncation $(I - P)\nabla F(W)$. With heterogeneous data or adapters, however, distinct $\{P_k\}$ mix gradients from different subspaces. Writing $g_k = \nabla F(W) + \delta_k$, and

$$\mathcal{B}_{\mathrm{proj}} = \underbrace{\Big( \frac{1}{K} \sum_{k=1}^{K} P_k - I \Big) \nabla F(W)}_{\text{loss of global directions}} + \underbrace{\frac{1}{K} \sum_{k=1}^{K} P_k\, \delta_k}_{\text{heterogeneity amplification}} .$$

The first term captures structural loss: global gradient components that are consistently dropped by most subspaces. The second term shows why heterogeneity is amplified: even if $\sum_k \delta_k \approx 0$, the projected terms $\{P_k \delta_k\}$ do not cancel out, since each $P_k$ rotates deviations differently. As a result, client heterogeneity translates into disproportionately large residual perturbations in the low-rank parameter space.

## 2.3 ADVERSARIAL AMPLIFICATION

Let $g_k \in \mathbb{R}^d$ be client $k$'s vectorized gradient and $\mathcal{H}$ the set of honest clients with $\bar{g}_{\mathcal{H}} = \frac{1}{|\mathcal{H}|} \sum_{k \in \mathcal{H}} g_k$. LoRA restricts updates to an $r$-dimensional subspace $\mathcal{S}$ with projector $P_{\mathcal{S}}$. Define the relative influence rate (RIR) of an adversarial gradient $g_{\mathrm{adv}}$ by

$$\mathrm{RIR} = \frac{\|P_{\mathcal{S}}(g_{\mathrm{adv}})\|}{\|P_{\mathcal{S}}(\bar{g}_{\mathcal{H}})\|}.$$

Assume honest gradients decompose as $g_k = \mu + \epsilon_k$ with $\mathbb{E}[\epsilon_k] = 0$ and $\mathrm{Cov}(\epsilon_k) = \sigma^2 I_d$. Then

$$\mathbb{E}\big[\|P_{\mathcal{S}}(\bar{g}_{\mathcal{H}})\|^2\big] = \|P_{\mathcal{S}}\mu\|^2 + \frac{\sigma^2 r}{|\mathcal{H}|}. \tag{3}$$

For a typical low-rank adapter, $\mathcal{S}$ is not aligned with $\mu$; under random orientation, $\mathbb{E}[\|P_{\mathcal{S}}\mu\|^2] = (r/d)\|\mu\|^2$, so when $r \ll d$ the variance term dominates, and

$$\|P_{\mathcal{S}}(\bar{g}_{\mathcal{H}})\| \approx \sigma\sqrt{\tfrac{r}{|\mathcal{H}|}}.$$

An adversary can align $g_{\mathrm{adv}}$ with $\mathcal{S}$, giving $\|P_{\mathcal{S}}(g_{\mathrm{adv}})\| \approx \sigma$. Hence

$$\mathrm{RIR}_{\mathrm{LoRA}} \approx \sqrt{\tfrac{|\mathcal{H}|}{r}}, \qquad \mathrm{RIR}_{\mathrm{full}} \approx \sqrt{\tfrac{|\mathcal{H}|}{d}}, \quad \Rightarrow \quad \frac{\mathrm{RIR}_{\mathrm{LoRA}}}{\mathrm{RIR}_{\mathrm{full}}} \approx \sqrt{\tfrac{d}{r}}.$$

Thus, adversarial influence is amplified by $\Theta(\sqrt{d/r})$ in LoRA FL relative to full-rank FL. Assumptions and proof of Equation 3 are detailed in Appendix B.

## 2.4 JACOBIAN SENSITIVITY OF LORA PARAMETERIZATION

A distinctive vulnerability of LoRA-based federated learning lies in the Jacobian structure induced by its bilinear adapter mapping. For an input $x$ and adapters $(A, B)$, the output of a LoRA-augmented layer is

$$z(x; A, B) = x\Big(W_0 + \tfrac{\alpha}{r}BA\Big).$$

Unlike full-rank fine-tuning, this output depends bilinearly on $(A, B)$. As a result, even small perturbations in either $A$ or $B$ can produce amplified changes in $z$ and in the downstream gradients. The amplification factor is proportional to the feature norm $\|x\|$, the spectral norms of $A$ and $B$, and the product of downstream spectral norms. Consequently:

- **Sensitivity to small perturbations.** A tiny change in $A$ or $B$ can be magnified if $A$ or $B$ is ill-conditioned or if $x$ has large energy.

- **Gradient instability.** The loss gradient with respect to $(A, B)$ is not only scaled by the Jacobian above, but also by the inverse temperature $1/T$ of the softmax. Hence, highly confident or adversarial logits can cause large swings in gradient updates.

- **Amplification of heterogeneity.** Clients with slightly different low-rank subspaces may project their updates into nearly orthogonal directions. Unlike full-rank training where heterogeneity cancels in expectation, here it can accumulate, leading to extreme update dispersion.

In summary, the Jacobian structure of LoRA explains why federated LoRA fine-tuning is more sensitive to both non-IID data and malicious perturbations than standard FL. We provide supporting derivations in Appendix C.

## 3 FRAMEWORK AND GLOBAL ALGORITHM

Motivated by the limitations in Section 2, we propose Robust Federated Distillation for LoRA (RFD-LoRA), which aggregates in logit space instead of averaging adapter parameters. Clients transmit soft labels on a reference dataset, and the server aggregates and distills them into the global model. This eliminates aggregation bias and initialization lag, bounds adversarial influence, and reduces non-IID sensitivity. We next formalize the training protocol.

### 3.1 CLIENT-SIDE PROCEDURE

Each client $k$ holds a private dataset $\mathcal{D}_k$ and a local LoRA adapter $(A_k, B_k)$ trained on top of the frozen base model $W_0$. Given a small reference dataset $\mathcal{D}_{\mathrm{ref}}$, which may be public or synthetically generated, the client computes logits

$$z_k(x) = (W_0 + \tfrac{\alpha}{r}B_k A_k)(x), \quad x \in \mathcal{D}_{\mathrm{ref}}. \tag{4}$$

**Input:** Base model $W_0$ (frozen or partially trainable); public anchor set $\mathcal{D}_{\text{ref}}$; clients $\{1..K\}$ with private
data $\{\mathcal{D}_k\}$; heterogeneous LoRA ranks $\{r_k\}$ and adapter structures; rounds $N$, local steps $E$;
client LR $\eta$, server LR $\gamma$; MoM groups $M$; clipping radius $c$; robustness hyperparams: $T_0, \kappa, \tau$
(CAT), $\lambda$ (MMD), $\rho$ (DIS).
**Output:** Global model $W$.
$W \leftarrow W_0$.
**for** $n \leftarrow 1$ **to** $N$ **do**
    // Client-side local fine-tuning (in parallel for $k=1..K$)
    **for** $k \in \{1..K\}$ *(in parallel)* **do**
        Initialize/continue local adapters $(A_k, B_k)$ with rank $r_k$ (no constraint across clients).
        **for** $e \leftarrow 1$ **to** $E$ **do**
            Sample $(x,y) \sim \mathcal{D}_k$; compute logits $z_k(x) = x\left(W + \frac{\alpha}{r_k} B_k A_k\right)$.
            Compute task loss $\ell_k$ and update $(A_k, B_k) \leftarrow (A_k, B_k) - \eta \nabla_{A_k, B_k} \ell_k$.
        **end**
        // Release clipped logits on anchors
        **for** $x \in \mathcal{D}_{ref}$ **do**
            send $z_k(x) \leftarrow \text{clip}(z_k(x), [-c, c])$ to server.
        **end**
    **end**
    // Server-side robust aggregation on anchors
    **for** $x \in \mathcal{D}_{ref}$ **do**
        Randomly partition clients into $M$ groups $\{G_m\}_{m=1}^M$ (Median-of-Means).
        **Group means:** $\bar{z}_m(x) \leftarrow \frac{1}{|G_m|} \sum_{k \in G_m} z_k(x)$;    $\overline{z^2}_m(x) \leftarrow \frac{1}{|G_m|} \sum_{k \in G_m} z_k(x)^{\circ 2}$.
        **MoM consensus:** $\tilde{z}(x) \leftarrow \text{median}\left(\{\bar{z}_m(x)\}_{m=1}^M\right)$.
        **Robust moments for MMD:** $\widehat{\mu}(x) \leftarrow \text{median}(\{\bar{z}_m(x)\})$,
        $\widehat{\sigma}^2(x) \leftarrow \text{median}(\{\overline{z^2}_m(x)\}) - \widehat{\mu}(x)^{\circ 2}$.
        // DIS: disagreement suppression (non-IID guard)
        $v(x) \leftarrow \sum_i \text{Var}_m(\bar{z}_m^{(i)}(x))$; $w(x) \leftarrow (1 + \rho\, v(x))^{-1}$.
        // CAT: confidence-adaptive temperature (stabilize gradients)
        $\tilde{q}(x) \leftarrow \text{softmax}(\tilde{z}(x))$; $T(x) \leftarrow T_0\left(1 + \kappa\left[\max_i \tilde{q}_i(x) - \tau\right]_+\right)$.
        // KD + MMD-KD objective (per-anchor)
        Student logits $z_W(x)$ from current $W$.
        $L_{\text{KD}}(x) \leftarrow \text{CE}\left(\text{softmax}(\tilde{z}(x)/T(x)), \text{softmax}(z_W(x)/T(x))\right)$.
        $L_{\text{MMD}}(x) \leftarrow \lambda\left(\|z_W(x) - \widehat{\mu}(x)\|_2^2 + \|(z_W(x) - \widehat{\mu}(x))^{\circ 2} - \widehat{\sigma}^2(x)\|_2^2\right)$.
        $L_{\text{RFD}}(x) \leftarrow w(x)\left(L_{\text{KD}}(x) + L_{\text{MMD}}(x)\right)$.
    **end**
    // Server update by distillation over anchors
    $W \leftarrow W - \gamma \cdot \nabla_W \left(\frac{1}{|\mathcal{D}_{\text{ref}}|} \sum_{x \in \mathcal{D}_{\text{ref}}} L_{\text{RFD}}(x)\right)$.
**end**
**return** $W$.

**Algorithm 1:** Training protocol of RFD-LoRA.

The logits are optionally clipped to ensure bounded energy,

$$\hat{z}_k(x) = \text{clip}(z_k(x), [-c, c]),$$

and converted into soft labels via temperature scaling

$$q_k(x) = \text{softmax}(\hat{z}_k(x)/T).$$

To mitigate over-confidence, we later introduce an adaptive temperature schedule (CAT). Finally, the client transmits $\{q_k(x)\}_{x \in \mathcal{D}_{\text{ref}}}$ to the server.

### 3.2 SERVER-SIDE AGGREGATION

Upon receiving predictions from all clients, the server aggregates them robustly to obtain a consensus distribution $\tilde{q}(x)$ for each $x \in \mathcal{D}_{\text{ref}}$. We adopt *Median-of-Means (MoM)* or coordinate-wise median, which are known to tolerate an $\varepsilon$-fraction of Byzantine clients. Formally, we show in Section 4.3 that

$$\|\tilde{q}(x) - \bar{q}_{\mathcal{H}}(x)\|_1 \leq O\left(\sqrt{\frac{1}{K}} + \sqrt{\varepsilon}\right),$$

where $\bar{q}_{\mathcal{H}}$ is the average distribution of honest clients. The aggregated distribution $\tilde{q}(x)$ then serves as the teacher for the global student model, updated by minimizing the distillation loss

$$\mathcal{L}_{\mathrm{KD}}(x) = \mathrm{KL}\big(\tilde{q}(x) \,\|\, p_W(x)\big),$$

where $p_W(x)$ denotes the global model's predictive distribution. In Section 4, we further enhance this objective with robustness modules, including Confidence-Adaptive Temperature (CAT), MMD-based Knowledge Distillation (MMD-KD), and Disagreement Suppression (DIS). We show the detailed training protocol in Algorithm 1.

## 4 ROBUSTNESS MODULES

### 4.1 CONFIDENCE-ADAPTIVE TEMPERATURE (CAT)

Given a predicted distribution $q_k(x)$ from client $k$, we define the adaptive temperature

$$T(x) = T_0\Big(1 + \kappa \cdot [\max_i \tilde{q}_i(x) - \tau]_+\Big), \tag{5}$$

where $T_0 \geq 1$ is a base temperature, $\kappa \geq 0$ is a scaling factor, $\tau \in [1/C, 1]$ is a confidence threshold, and $[u]_+ = \max(u, 0)$. The student distribution is computed as

$$p_W(x) = \mathrm{softmax}\big(z_W(x)/T(x)\big).$$

**Theorem 1 (Gradient sensitivity under CAT).** Let $J_W(x) = \partial z_W(x)/\partial W$ denote the Jacobian of logits. Then for any input $x$, the update step satisfies

$$\|\nabla_W \mathcal{L}_{\mathrm{KD}}(x)\| \leq \frac{C}{T(x)}\|J_W(x)\|, \tag{6}$$

where $C$ is a constant depending only on the clipping bound $c$ and the number of classes $C$.

*Proof sketch.* The derivative of $\mathrm{softmax}(z/T)$ w.r.t. $z$ has operator norm at most $1/(4T)$, hence the difference $\|p_W(x) - \tilde{q}(x)\|_2$ is $\mathcal{O}(1/T)$. Combining with the chain rule $\nabla_W = J_W^\top \nabla_z$ yields the bound. A full proof is given in Appendix D.

### 4.2 MMD-BASED KNOWLEDGE DISTILLATION (MMD-KD)

To mitigate energy-based manipulations where adversaries distort logit magnitudes, we align both first- and second-order logit statistics via Maximum Mean Discrepancy (MMD). Let the server aggregate both the mean $\hat{\mu}(x)$ and diagonal variance $\widehat{\sigma}^2(x)$ of logits across clients:

$$\hat{\mu}(x) = \frac{1}{K}\sum_k \hat{z}_k(x), \quad \widehat{\sigma}^2(x) = \frac{1}{K}\sum_k \big(\hat{z}_k(x) - \hat{\mu}(x)\big)^2.$$

We define the MMD-KD loss as

$$\mathcal{L}_{\mathrm{MMD}}(x) = \lambda\Big(\|z_W(x) - \hat{\mu}(x)\|_2^2 + \|\mathrm{Var}[z_W(x)] - \widehat{\sigma}^2(x)\|_2^2\Big), \tag{7}$$

where $\lambda > 0$ is a regularization coefficient.

**Theorem 2 (Variance-constrained robustness).** Assume clipped logits are $\sigma$-sub-Gaussian across honest clients. Then with probability at least $1 - \delta$, the aggregated variance satisfies

$$\|\widehat{\sigma}^2(x) - \sigma^2(x)\|_\infty \leq O\Big(\sqrt{\tfrac{\log(1/\delta)}{K}} + \sqrt{\varepsilon}\Big).$$

Consequently, the additional MMD-KD term bounds the adversarial amplification due to logit energy distortions by at most

$$\|\nabla_W \mathcal{L}_{\mathrm{MMD}}(x)\| \leq \lambda C_{\mathrm{mmd}} \cdot \|J_W(x)\| \cdot \Big(\sqrt{\tfrac{1}{K}} + \sqrt{\varepsilon}\Big). \tag{8}$$

*Proof sketch.* Concentration inequalities for sub-Gaussian random variables give uniform convergence of the variance estimate under MoM aggregation. The gradient bound follows from applying Lipschitz continuity of the squared loss and chain rule. Full details are provided in Appendix E.

Table 1: Adversarial robustness on GLUE (average over MNLI-m/mm, SST-2, QQP, QNLI). Poisoned-client fraction $\rho \in \{0.10, 0.30\}$; per-poisoned-client poison rate $\pi = 0.20$ (trigger-token insertion, fixed target label). CA = clean accuracy on clean test inputs; RA = accuracy on attacked inputs; ASR (lower is better) = targeted success on triggered inputs. LoRA rank = 8 for baselines; FD-LoRA aggregates logits and supports heterogeneous ranks. Means over 5 runs.

(a) IID clients

| Method | CA | $\rho = 0.10$ | | $\rho = 0.30$ | |
|---|---|---|---|---|---|
| | | RA | ASR↓ | RA | ASR↓ |
| FedAvg | 88.2 | 84.0 | 18.5 | 78.3 | 35.2 |
| FFA-LoRA | 88.9 | 86.2 | 12.3 | 81.5 | 24.8 |
| FedIT | 89.0 | 86.0 | 12.7 | 81.2 | 25.1 |
| FLoRA | 89.2 | 86.5 | 11.9 | 81.7 | 23.7 |
| FlexLoRA | 89.7 | 87.2 | 9.8 | 83.0 | 19.6 |
| LoRA-Fair | 89.7 | 87.5 | 9.5 | 83.3 | 18.9 |
| **RFD-LoRA** | **90.6** | **89.1** | **5.1** | **86.8** | **10.7** |

(b) Severe non-IID clients

| Method | CA | $\rho = 0.10$ | | $\rho = 0.30$ | |
|---|---|---|---|---|---|
| | | RA | ASR↓ | RA | ASR↓ |
| FedAvg | 86.1 | 80.2 | 22.4 | 72.0 | 36.7 |
| FFA-LoRA | 88.0 | 84.8 | 14.2 | 78.3 | 27.9 |
| FedIT | 87.9 | 84.5 | 14.6 | 78.0 | 28.2 |
| FLoRA | 88.0 | 84.7 | 14.0 | 78.4 | 27.5 |
| FlexLoRA | 88.5 | 86.0 | 11.3 | 80.5 | 21.1 |
| LoRA-Fair | 88.6 | 86.2 | 10.9 | 80.8 | 20.4 |
| **RFD-LoRA** | **90.0** | **88.5** | **6.4** | **85.0** | **12.9** |

## 4.3 DISAGREEMENT SUPPRESSION (DIS)

To suppress the effect of non-IID clients, we compute the group variance of aggregated predictions. Partition the $K$ clients into $M$ groups and let

$$\bar{q}_m(x) = \frac{1}{|G_m|} \sum_{k \in G_m} q_k(x), \quad v(x) = \sum_{i=1}^{C} \mathrm{Var}_m\big[\bar{q}_m^{(i)}(x)\big].$$

The sample weight is defined as

$$w(x) = \frac{1}{1 + \rho\, v(x)}, \quad \rho \geq 0. \tag{9}$$

The distillation loss is reweighted as

$$\mathcal{L}_{\mathrm{DIS}}(x) = w(x) \cdot \mathcal{L}_{\mathrm{KD}}(x).$$

**Theorem 3 (Variance-adaptive error bound).** Suppose the expected group variance of honest clients satisfies $\mathbb{E}[v_{\mathcal{H}}(x)] \leq H$. Then under MoM aggregation, the expected deviation is bounded by

$$\mathbb{E}_x\big[\|\tilde{q}_w(x) - \bar{q}_{\mathcal{H},w}(x)\|_1\big] \leq O\Big(\sqrt{\tfrac{H}{K}} + \sqrt{\varepsilon}\Big), \tag{10}$$

where $\bar{q}_{\mathcal{H},w}$ is the weighted average of honest client predictions.

*Proof sketch.* The weighting scheme ensures $\mathbb{E}[w(x)^2 v(x)] \leq O(H)$, which reduces the effective variance in concentration bounds for MoM. The adversarial contribution remains $O(\sqrt{\varepsilon})$. Full proof is given in Appendix F.

## 5 EXPERIMENTS

In this section, we evaluate RFD-LoRA through experiments. Results show that it achieves stronger robustness than existing federated LoRA methods under non-IID distributions and adversarial attacks, while also outperforming baselines in IID settings. Full experimental details, including datasets, model configurations, and hyperparameters, are in Appendix G and H.

### 5.1 EXPERIMENTAL RESULTS

Table 1 summarizes adversarial robustness under targeted backdoor attacks with varying fractions of poisoned clients. RFD-LoRA consistently achieves the highest clean accuracy (CA) across both IID and severe non-IID settings, confirming that robustness does not come at the expense of standard performance. More importantly, under adversarial conditions, RFD-LoRA yields substantially higher robust accuracy (RA) and lower attack success rate (ASR) than all baselines. In the IID case,

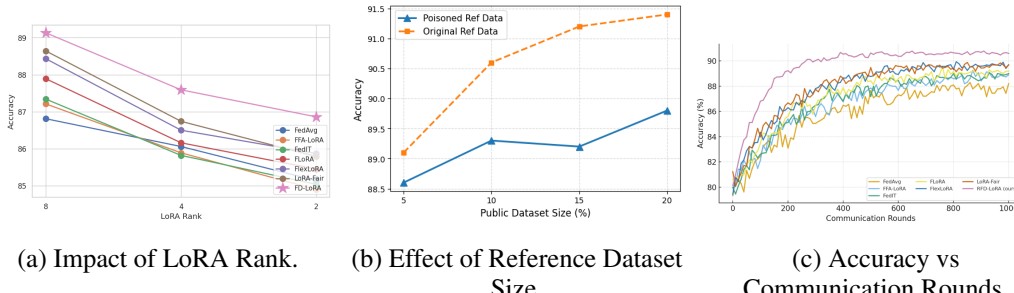

(a) Impact of LoRA Rank.    (b) Effect of Reference Dataset Size.    (c) Accuracy vs Communication Rounds.

Figure 1: RFD-LoRA's performance across various rank, reference dataset size, and training round.

Table 2: Comparison of server communication cost for different state-of-the-art approaches.

| Method | Param Size Per Round | Rank-dependent Cost |
|---|---|---|
| FFA-LoRA | 1.6MB | Yes |
| FedIT | 3.2MB | Yes |
| FLoRA | 9.5MB | Yes |
| FlexLoRA | 6.4MB | Yes |
| LoRA-Fair | 3.6MB | Yes |
| **RFD-LoRA** | **12KB** | **No** |

with $\rho = 0.10$ poisoned clients, RFD-LoRA achieves RA of 89.1 and ASR of only 5.1, while the best baseline (LoRA-Fair) reaches 87.5 RA and 9.5 ASR. At $\rho = 0.30$, the gap widens: RFD-LoRA maintains 86.8 RA with 10.7 ASR, compared to LoRA-Fair's 83.3 RA and 18.9 ASR. Similar trends appear under severe non-IID distributions: with $\rho = 0.30$, FedAvg and FFA-LoRA suffer ASR above 27%, while RFD-LoRA cuts ASR to 12.9% and preserves 85.0% RA. These results show that while all methods degrade as $\rho$ increases, RFD-LoRA degrades much more slowly, indicating better resilience. Overall, the results validate the effectiveness of our three robustness modules CAT, MMD-KD, and DIS in suppressing adversarial amplification and mitigating non-IID drift. RFD-LoRA not only provides stronger defense against poisoned clients but also achieves superior clean-task accuracy compared to existing federated LoRA approaches.

Moreover, Table 2 shows the server communication cost per round. While existing federated LoRA methods require transmitting millions of parameters and their cost scales with adapter rank, RFD-LoRA only uploads logits, reducing the per-round size to just 12KB. This rank-independent design eliminates the need to synchronize full adapter weights, saving both memory and computation while enabling practical deployment across heterogeneous clients.

## 5.2 ABLATION STUDY

**Robustness modules.** To isolate the effect of each robustness module in RFD-LoRA, we conduct ablation studies by removing CAT, MMD-KD, or DIS individually. Results in the left of Table 3 under both IID and severe non-IID partitions show that eliminating any one component consistently reduces performance, confirming their complementary roles. Removing CAT leads to unstable training and larger variance across runs, highlighting its role in controlling Jacobian sensitivity. Without MMD-KD, the model becomes vulnerable to energy-shaping attacks, as indicated by sharper performance drops under adversarial clients. Finally, disabling DIS amplifies the effect of non-IID heterogeneity, producing significant degradation. Together, these results validate that the three modules jointly contribute to the robustness of RFD-LoRA, and that each component addresses a distinct failure mode of federated LoRA.

**Robust aggregation.** The right side of Table 3 compares different server-side aggregators under $\rho=0.30$ poisoned clients. Simple averaging performs worst, while coordinate-wise median offers partial robustness. Our Median-of-Means (MoM) aggregator achieves the highest clean accuracy and robust accuracy, and reduces ASR by more than half compared to mean aggregation. We also vary the number of MoM groups $M$ and find that $M=5$ yields the lowest ASR, whereas very small

Table 3: Ablations on RFD-LoRA modules (left) and robust aggregation (right). Top right: aggregator choice under $\rho$=0.30. Bottom right: impact of varying MoM group count $M$.

| Variant | IID | | | Severe non-IID | | |
|---------|-----|-----|------|----------------|-----|------|
| | CA | RA | ASR↓ | CA | RA | ASR↓ |
| Full | **90.6** | **89.1** | **5.1** | **90.0** | **88.5** | **6.4** |
| w/o CAT | 90.1 | 87.3 | 7.9 | 89.3 | 85.6 | 10.2 |
| w/o MMD-KD | 90.4 | 86.8 | 9.8 | 89.5 | 85.1 | 13.7 |
| w/o DIS | 90.3 | 86.2 | 8.7 | 88.4 | 83.0 | 15.1 |
| CAT only | 89.9 | 87.0 | 8.2 | 88.7 | 84.2 | 11.9 |
| MMD-KD only | 90.0 | 86.5 | 10.5 | 88.9 | 84.0 | 14.5 |
| DIS only | 89.8 | 87.6 | 8.4 | 89.1 | 86.3 | 11.2 |

| Aggregator Choice ($\rho$=0.30) | | | |
|---------|------|------|------|
| Method | CA | RA | ASR↓ |
| Mean | 89.6 | 83.1 | 22.8 |
| Coord. Median | 89.7 | 84.9 | 16.2 |
| MoM (ours) | **90.6** | **86.8** | **10.7** |
| MoM Groups $M$ (ASR↓) | | | |
| $M = 3$ | 11.9 | | |
| $M = 5$ | **10.7** | | |
| $M = 7$ | 11.0 | | |
| $M = 9$ | 12.5 | | |

Table 4: Ablations on (a) CAT schedule, (b) MMD-KD strength, and (c) DIS weighting. Metrics averaged over GLUE tasks under $\rho$=0.30 poisoned clients. CA = clean accuracy; RA = robust accuracy; ASR (lower is better).

(a) CAT schedule

| $(T_0, \kappa, \tau)$ | CA | RA | ASR↓ |
|-----------------------|------|------|------|
| (1 , 0 , 0.7) | 90.2 | 85.5 | 14.8 |
| (2 , 2 , 0.7) | **90.6** | **86.8** | **10.7** |
| (3 , 4 , 0.7) | 90.0 | 86.2 | 12.5 |
| (2 , 2 , 0.6) | 90.4 | 86.5 | 11.3 |
| (2 , 2 , 0.8) | 90.1 | 86.0 | 12.0 |
| (4 , 4 , 0.8) | 89.7 | 85.6 | 14.3 |

(b) MMD-KD strength

| Variant | CA | RA | ASR↓ |
|---------|------|------|------|
| $\lambda = 0$ | 90.3 | 85.0 | 15.6 |
| $\lambda = 0.05$ | 90.5 | 86.2 | 12.4 |
| $\lambda = 0.1$ | **90.6** | **86.8** | **10.7** |
| $\lambda = 0.2$ | 90.1 | 86.0 | 12.8 |
| Mean-only | 90.4 | 86.3 | 11.9 |
| Mean+Var | **90.6** | **86.8** | **10.7** |

(c) DIS weighting

| Variant | CA | RA | ASR↓ |
|---------|------|------|------|
| $\rho = 0$ | 90.3 | 86.2 | 13.7 |
| $\rho = 0.5$ | 90.5 | 86.6 | 11.5 |
| $\rho = 1.0$ | **90.6** | **86.8** | **10.7** |
| $\rho = 2.0$ | 90.0 | 86.0 | 12.6 |
| Client-var | 90.2 | 86.1 | 12.9 |
| Group-var | **90.6** | **86.8** | **10.7** |

or very large $M$ slightly degrade robustness due to under- or over-fragmentation. This confirms that MoM provides a strong and stable defense for logit aggregation in federated settings.

**Hyperparameter settings.** Table 4 evaluates the effect of hyperparameters tuning in each module. For CAT, moderate temperature and confidence scaling ($T_0$=2, $\kappa$=2, $\tau$=0.7) yields the best tradeoff, improving RA while significantly lowering ASR; overly large schedules degrade CA due to over-softening. For MMD-KD, introducing moment alignment ($\lambda$=0.1) provides clear gains over plain KD, and matching both mean and variance further reduces ASR without hurting CA. Finally, DIS weighting improves robustness as $\rho$ increases, with $\rho$=1.0 giving the best balance; group-variance estimation is consistently superior to client-variance, confirming its stability under poisoned clients.

**Other factors.** We analyze factors shaping RFD-LoRA's performance. Figure 1(a) shows that while accuracy declines with lower LoRA ranks, RFD-LoRA consistently outperforms baselines, demonstrating robustness to low-dimensional adaptation. Figure 1(b) evaluates reference data: larger anchor sets improve accuracy but plateau near 20%. When 30% of the reference set is poisoned by trigger-token insertion and label flipping, RFD-LoRA remains relatively stable, though accuracy drops compared to clean data—highlighting the importance of anchor quality. Figure 1(c) plots accuracy over communication rounds: competing methods converge slowly with oscillations, while RFD-LoRA stabilizes quickly and achieves the highest final accuracy.

## 6 CONCLUSION

We introduced RFD-LoRA, the first federated distillation framework for LoRA fine-tuning. Our analysis exposes core weaknesses of federated LoRA: aggregation, initialization and projection bias, adversarial amplification, and Jacobian sensitivity. RFD-LoRA aggregates in logit space, eliminating parameter-space bias and enabling heterogeneous client ranks/adapter designs. With Confidence-Adaptive Temperature (CAT), MMD-based Distillation (MMD-KD), and Disagreement Suppression (DIS), we provide error bounds and show these modules directly reduce amplification and sensitivity while improving robustness under non-IID and adversarial clients. Experiments on GLUE confirm consistent gains in accuracy, robustness, and communication efficiency over prior federated LoRA methods, suggesting a practical path to robust, parameter-efficient FL.

## 7 REPRODUCIBILITY STATEMENT

We have made extensive efforts to ensure reproducibility of our results. The detailed training setup, including dataset descriptions, partitioning strategies, and hyperparameters, is provided in Appendix G. All algorithms are fully specified in Algorithm 1, with theoretical assumptions and complete proofs included in Appendix B - F. Experimental protocols, evaluation metrics, and ablation studies are reported in Appendix G and H. An anonymous implementation of RFD-LoRA, including preprocessing scripts and training code, has been uploaded to supplementary material to facilitate exact reproduction of our findings.

## 8 USE OF LARGE LANGUAGE MODELS (LLMS)

In this manuscript, we made limited use of a large language model (LLM) solely for writing-related assistance. Specifically, the LLM was employed to help with drafting, rephrasing, and polishing text for improved clarity and readability. All technical content, including the design of methods, derivation of theorems, mathematical proofs, and experimental design and execution, was fully conceived, implemented, and validated by the authors. No LLM was used for generating data, conducting experiments, analyzing results, or developing the core scientific contributions of this work. The responsibility for all ideas, technical claims, and conclusions lies entirely with the authors.

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

## A  INITIALIZATION LAG

In frameworks such as FLoRA (Wang et al. (2024)), the local matrices $A_k$ and $B_k$ are re-sampled each round, with $A_k \sim \mathcal{N}(0, \sigma^2)$ and $B_k$ set to zero. Although this design guarantees that the reconstructed global model $W_0 + \Delta W'$ remains aligned with the base model $W_0$, it induces what we callclient-side initialization lag. In particular, freshly reinitialized adapters yield ill-conditioned gradients during the first few local steps. For a forward pass with LoRA-modified weights

$$y = x(W_0 + B_k A_k), \tag{11}$$

the gradients of $A_k$ and $B_k$ with respect to the loss $L$ are

$$\frac{\partial L}{\partial A_k} = x^\top \frac{\partial L}{\partial y} B_k^\top, \qquad \frac{\partial L}{\partial B_k} = A_k^\top x^\top \frac{\partial L}{\partial y}. \tag{12}$$

At initialization, since $B_k = 0$ and $A_k$ is random, we obtain

$$\frac{\partial L}{\partial A_k} \to 0, \qquad \frac{\partial L}{\partial B_k} \to \text{random direction.}$$

As a result, clients waste many updates simply overcoming the poor initialization, leading to inefficient early-stage training.

## B  ADVERSARIAL AMPLIFICATION ANALYSIS

**Lemma I.** Let $\{\epsilon_k\}_{k\in\mathcal{H}}$ be i.i.d. random vectors in $\mathbb{R}^d$ with $\mathbb{E}[\epsilon_k] = 0$ and $\text{Cov}(\epsilon_k) = \sigma^2 I_d$. Let $P_\mathcal{S} \in \mathbb{R}^{d\times d}$ be the orthogonal projector onto an $r$-dimensional subspace $\mathcal{S}$ (i.e., $P_\mathcal{S} = P_\mathcal{S}^\top = P_\mathcal{S}^2$ and $\text{tr}(P_\mathcal{S}) = r$). Define the sample mean $\bar{\epsilon} = \frac{1}{|\mathcal{H}|} \sum_{k\in\mathcal{H}} \epsilon_k$. Then

$$\mathbb{E}\big[\|P_\mathcal{S}\bar{\epsilon}\|^2\big] = \frac{\sigma^2 r}{|\mathcal{H}|}.$$

**Proof:** Since $\bar{\epsilon}$ has mean 0 and $\text{Cov}(\bar{\epsilon}) = \frac{1}{|\mathcal{H}|} \text{Cov}(\epsilon_k) = \frac{\sigma^2}{|\mathcal{H}|} I_d$, we compute

$$\mathbb{E}\big[\|P_\mathcal{S}\bar{\epsilon}\|^2\big] = \mathbb{E}\big[(P_\mathcal{S}\bar{\epsilon})^\top (P_\mathcal{S}\bar{\epsilon})\big] = \mathbb{E}\big[\bar{\epsilon}^\top P_\mathcal{S}^\top P_\mathcal{S}\bar{\epsilon}\big].$$

Because $P_\mathcal{S}$ is an orthogonal projector matrix, it is both symmetric ($P_\mathcal{S}^\top = P_\mathcal{S}$) and idempotent ($P_\mathcal{S}^2 = P_\mathcal{S}$). Hence $P_\mathcal{S}^\top P_\mathcal{S} = P_\mathcal{S} P_\mathcal{S} = P_\mathcal{S}$, and therefore

$$\mathbb{E}\big[\bar{\epsilon}^\top P_\mathcal{S}^\top P_\mathcal{S}\bar{\epsilon}\big] = \mathbb{E}\big[\bar{\epsilon}^\top P_\mathcal{S}\bar{\epsilon}\big].$$

Using the identity $\mathbb{E}[x^\top A x] = \text{tr}(A \text{Cov}(x))$ for zero-mean $x$, we obtain

$$\mathbb{E}\big[\bar{\epsilon}^\top P_\mathcal{S}\bar{\epsilon}\big] = \text{tr}\big(P_\mathcal{S} \text{Cov}(\bar{\epsilon})\big) = \frac{\sigma^2}{|\mathcal{H}|} \text{tr}(P_\mathcal{S}) = \frac{\sigma^2}{|\mathcal{H}|} r,$$

since the trace of an orthogonal projector equals its rank. This completes the proof.

## C  JACOBIAN SENSITIVITY: SKETCHES OF ANALYSIS

We give rough arguments supporting the claims here:

**First-order sensitivity.**  Expanding $z(x; A + \Delta A, B + \Delta B)$ shows

$$\Delta z \approx \frac{\alpha}{r} x(B\Delta A + \Delta B A),$$

ignoring higher-order terms. Hence

$$\|\Delta z\| \lesssim \frac{\alpha}{r} \|x\|\big(\|B\|\|\Delta A\| + \|A\|\|\Delta B\|\big).$$

This shows that perturbations are scaled by $\|x\|$ and the adapter norms.

**Gradient sensitivity.** For softmax cross-entropy, the gradient with respect to logits satisfies

$$\|\nabla_z \ell(z)\| \leq \tfrac{2}{T}, \qquad \|\nabla_z \ell(z) - \nabla_z \ell(z')\| \leq \tfrac{1}{T}\|z - z'\|.$$

Thus gradient variation inherits the same amplification factor as $\Delta z$, multiplied by $1/T$. This explains why small adapter perturbations can trigger large swings when $T$ is small or when clients produce highly confident predictions.

**Deep network composition.** If the LoRA block lies at layer $\ell$, then downstream spectral norms $\rho_j$ further multiply the sensitivity, giving a bound on the order of

$$\frac{\alpha}{r}\|h_{\ell-1}\|(\|A\| + \|B\|)\prod_{j>\ell}\rho_j.$$

**Client heterogeneity amplification.** Two clients $(A_1, B_1)$ and $(A_2, B_2)$ yield different logits:

$$\|z(x; A_1, B_1) - z(x; A_2, B_2)\| \lesssim \tfrac{\alpha}{r}\|x\|\big(\|B_1\|\|A_1 - A_2\| + \|A_2\|\|B_1 - B_2\|\big).$$

If their low-rank subspaces are misaligned, the differences may not cancel but instead compound, unlike full-rank training.

**Summary.** The Jacobian analysis shows that LoRA adapters create scaling factors—through $\|x\|$, $\|A\|, \|B\|$, inverse temperature $1/T$, and downstream spectral norms—that magnify small differences. This accounts for the observed instability of federated LoRA under non-IID data and adversarial manipulation.

## D  GRADIENT SENSITIVITY UNDER CAT

**Theorem 1 (Gradient sensitivity under CAT).** Let $x$ be any input and let the student distribution be

$$p_W(x) = \text{softmax}\big(z_W(x)/T(x)\big),$$

where $T(x) \geq 1$ is the confidence-adaptive temperature and $z_W(x) \in \mathbb{R}^C$ are the logits of the global model. Let $\tilde{q}(x) \in \Delta^{C-1}$ be the server-aggregated teacher distribution (obtained from client logits after clipping and temperature scaling). Consider the distillation loss

$$\mathcal{L}_{\text{KD}}(x) = \text{KL}\big(\tilde{q}(x) \,\|\, p_W(x)\big).$$

Denote by $J_W(x) = \partial z_W(x)/\partial W$ the Jacobian of logits with respect to parameters $W$, and let $\|\cdot\|$ be the spectral/operator norm for matrices and the Euclidean norm for vectors. Then

$$\big\|\nabla_W \mathcal{L}_{\text{KD}}(x)\big\| \leq \frac{\Gamma}{T(x)}\|J_W(x)\|, \qquad \text{with } \Gamma \leq 2. \tag{13}$$

If, in addition, both student and teacher logits are clipped coordinate-wise to $[-c, c]$ before the softmax with temperature $T(x)$, then a refined bound holds:

$$\big\|\nabla_W \mathcal{L}_{\text{KD}}(x)\big\| \leq \frac{\sqrt{C}\,(M_T - m_T)}{T(x)}\|J_W(x)\|, \tag{14}$$

where, writing $a := c/T(x)$,

$$m_T = \frac{1}{1 + (C-1)e^{2a}}, \qquad M_T = \frac{1}{1 + (C-1)e^{-2a}}, \qquad 0 < m_T \leq M_T < 1.$$

**Proof:** Write $u(x) := z_W(x)/T(x) \in \mathbb{R}^C$ so that $p_W(x) = \text{softmax}(u(x))$. By definition,

$$\mathcal{L}_{\text{KD}}(x) = \sum_{i=1}^{C} \tilde{q}_i(x)\,\log\frac{\tilde{q}_i(x)}{p_i(x)} = \text{const} - \sum_{i=1}^{C} \tilde{q}_i(x)\,\log p_i(x),$$

where the term $\sum_i \tilde{q}_i \log \tilde{q}_i$ does not depend on $W$. The gradient with respect to the *pre-softmax* variables $u$ is standard. Using $\partial \log p_i / \partial u_j = \delta_{ij} - p_j$ yields

$$\frac{\partial \mathcal{L}_{\text{KD}}}{\partial u_j} = -\sum_{i=1}^{C} \tilde{q}_i \left( \delta_{ij} - p_j \right) = -\tilde{q}_j + p_j \sum_{i=1}^{C} \tilde{q}_i = p_j - \tilde{q}_j.$$

Hence, by the chain rule from $z$ to $u$,

$$\nabla_z \, \mathcal{L}_{\text{KD}}(x) = \frac{1}{T(x)} \left( p_W(x) - \tilde{q}(x) \right). \tag{15}$$

Applying the chain rule from $z$ to $W$ with $J_W(x) = \partial z_W(x)/\partial W$,

$$\nabla_W \, \mathcal{L}_{\text{KD}}(x) = J_W(x)^\top \, \nabla_z \, \mathcal{L}_{\text{KD}}(x) = \frac{1}{T(x)} \, J_W(x)^\top \left( p_W(x) - \tilde{q}(x) \right).$$

Taking norms and using submultiplicativity gives

$$\left\| \nabla_W \, \mathcal{L}_{\text{KD}}(x) \right\| \leq \frac{1}{T(x)} \left\| J_W(x) \right\| \left\| p_W(x) - \tilde{q}(x) \right\|_2. \tag{16}$$

For the coarse universal bound equation 13, note that $p_W(x)$ and $\tilde{q}(x)$ are probability vectors in the simplex. Thus $\|p_W(x) - \tilde{q}(x)\|_2 \leq \|p_W(x) - \tilde{q}(x)\|_1 \leq 2$. Plugging this into equation 16 yields equation 13 with $\Gamma = 2$.

For the refined bound equation 14, assume teacher and student logits are clipped to $[-c, c]$ before applying the softmax with temperature $T(x)$. Then each coordinate of $p_W(x)$ and $\tilde{q}(x)$ lies in the interval $[m_T, M_T]$, where the extrema follow from monotonicity and are attained at the corners of the hypercube:

$$m_T = \min_{z \in [-c,c]^C} \frac{e^{z_i/T(x)}}{\sum_j e^{z_j/T(x)}} = \frac{e^{-c/T(x)}}{e^{-c/T(x)} + (C-1)e^{c/T(x)}} = \frac{1}{1 + (C-1)e^{2a}},$$

$$M_T = \max_{z \in [-c,c]^C} \frac{e^{z_i/T(x)}}{\sum_j e^{z_j/T(x)}} = \frac{e^{c/T(x)}}{e^{c/T(x)} + (C-1)e^{-c/T(x)}} = \frac{1}{1 + (C-1)e^{-2a}},$$

with $a = c/T(x)$. Consequently, for every coordinate $i$, $|p_i(x) - \tilde{q}_i(x)| \leq M_T - m_T$, and therefore by Cauchy–Schwarz, $\|p_W(x) - \tilde{q}(x)\|_2 \leq \sqrt{C} \, (M_T - m_T)$. Substituting into equation 16 gives equation 14.

Both bounds are inversely proportional to $T(x)$, which shows that CAT reduces the gradient sensitivity by explicitly scaling the update with $1/T(x)$. This completes the proof.

## E  VARIANCE-CONSTRAINED ROBUSTNESS FOR MMD-KD

**Theorem 2 (Variance-constrained robustness for MMD-KD).** Fix an input $x$ and a class index set $\{1, \ldots, C\}$. For each client $k$, let the clipped logit vector be $\hat{z}_k(x) \in [-c, c]^C$. Assume the honest clients $\mathcal{H}$ (with $|\mathcal{H}| = (1 - \varepsilon)K$ and $\varepsilon < 1/2$) generate i.i.d. coordinates

$$\hat{z}_{k,i}(x) \text{ are } \sigma\text{-sub-Gaussian and bounded in } [-c, c], \qquad i = 1, \ldots, C.$$

Let $\mu_i = \mathbb{E}[\hat{z}_{k,i}(x)]$, $m_{2,i} = \mathbb{E}[\hat{z}_{k,i}(x)^2]$, and $\sigma_i^2 = m_{2,i} - \mu_i^2$ denote the honest mean, second moment, and variance, respectively. Construct Median-of-Means (MoM) estimators $\hat{\mu}_i$ and $\hat{m}_{2,i}$ using $M$ groups of size $b = K/M$ (group means, then coordinate-wise median across groups), and define

$$\hat{\sigma}_i^2 = \hat{m}_{2,i} - \hat{\mu}_i^2, \qquad \hat{\boldsymbol{\sigma}}^2 = (\hat{\sigma}_i^2)_{i=1}^C, \quad \boldsymbol{\sigma}^2 = (\sigma_i^2)_{i=1}^C.$$

Then there exist absolute constants $c_1, c_2, c_3 > 0$ such that, choosing $M \simeq c_1 \log(C/\delta)$,

$$\left\| \hat{\boldsymbol{\sigma}}^2 - \boldsymbol{\sigma}^2 \right\|_\infty \leq c_2 \, (c^2 + \sigma^2) \left( \sqrt{\frac{\log(C/\delta)}{K}} + \sqrt{\varepsilon} \right) \tag{17}$$

holds with probability at least $1 - \delta$. Consider the MMD-KD loss (linear-kernel surrogate) at $x$,

$$\mathcal{L}_{\text{MMD}}(x) = \lambda \Big( \|z_W(x) - \hat{\boldsymbol{\mu}}(x)\|_2^2 + \|(z_W(x) - \hat{\boldsymbol{\mu}}(x))^{\circ 2} - \hat{\boldsymbol{\sigma}}^2(x)\|_2^2 \Big), \tag{18}$$

where $(\cdot)^{\circ 2}$ denotes element-wise square and $z_W(x)$ are student logits with Jacobian $J_W(x) = \partial z_W(x)/\partial W$. If $\|z_W(x)\|_\infty \leq c$ (via clipping), then the excess gradient induced by aggregation error satisfies

$$\left\| \nabla_W \mathcal{L}_{\mathrm{MMD}}(x) - \nabla_W \mathcal{L}_{\mathrm{MMD}}^\star(x) \right\| \leq \lambda\, C_{\mathrm{mmd}}\, \|J_W(x)\| \left( \sqrt{\tfrac{\log(C/\delta)}{K}} + \sqrt{\varepsilon} \right), \qquad (19)$$

where $\mathcal{L}_{\mathrm{MMD}}^\star$ is the ideal loss with $(\widehat{\boldsymbol{\mu}}, \widehat{\boldsymbol{\sigma}}^2)$ replaced by $(\boldsymbol{\mu}, \boldsymbol{\sigma}^2)$, and $C_{\mathrm{mmd}}$ depends only on $c$, $\sigma$, and $C$.

**Proof:** The proof has three steps: robust mean and second-moment concentration under MoM, propagation to the variance estimator, and a stability bound for the gradient of equation 18.

**Step I: MoM concentration for mean and second moment.** Randomly partition the $K$ clients into $M$ groups $G_1, \ldots, G_M$ of size $b = K/M$ (assume $M$ divides $K$). For each coordinate $i$ define group means

$$\overline{Z}_{m,i} = \frac{1}{|G_m|} \sum_{k \in G_m} \hat{z}_{k,i}, \qquad \overline{U}_{m,i} = \frac{1}{|G_m|} \sum_{k \in G_m} \hat{z}_{k,i}^2,$$

and set the MoM estimators $\widehat{\mu}_i = \mathrm{median}\{\overline{Z}_{m,i}\}_{m=1}^M$ and $\widehat{m}_{2,i} = \mathrm{median}\{\overline{U}_{m,i}\}_{m=1}^M$.

Under the $\varepsilon$-contamination model with $\varepsilon < 1/2$, at least $(1-2\varepsilon)M$ groups contain an adversarial fraction at most $1/2$ (standard Chernoff-style argument for random partition; details omitted for brevity). For honest samples, $\hat{z}_{k,i}$ are $\sigma$-sub-Gaussian and bounded by $c$, hence each honest group mean $\overline{Z}_{m,i}$ is sub-Gaussian with parameter $\lesssim \sigma/\sqrt{b}$ and satisfies (by Hoeffding/Bernstein)

$$\Pr\left( |\overline{Z}_{m,i} - \mu_i| > t \right) \leq 2\exp\left( -c'\, b\, \min\{\tfrac{t^2}{\sigma^2}, \tfrac{t}{c}\} \right).$$

A coordinate-wise application and a union bound across $M$ groups imply that with probability at least $1 - \delta/2$, for all $i$ at least half of the groups satisfy

$$|\overline{Z}_{m,i} - \mu_i| \leq c'' \max\left\{ \sigma \sqrt{\tfrac{\log(CM/\delta)}{b}}, \; c\, \tfrac{\log(CM/\delta)}{b} \right\}.$$

Since $b = K/M$ and $M \simeq c_1 \log(C/\delta)$, the first term dominates, giving

$$\|\widehat{\boldsymbol{\mu}} - \boldsymbol{\mu}\|_\infty \leq c_3\, \sigma \sqrt{\tfrac{\log(C/\delta)}{K}} + c_4\, \sigma \sqrt{\varepsilon}. \qquad (20)$$

The $\sqrt{\varepsilon}$ term follows from the breakdown-point property of the coordinate-wise median: at most an $\varepsilon$-fraction of groups can be arbitrarily corrupted, and the median discards them up to a factor absorbed in constants.

For second moments, note that $\hat{z}_{k,i}^2 \in [0, c^2]$ are sub-exponential with parameter $\lesssim c^2$, hence the same MoM argument yields

$$\|\widehat{\boldsymbol{m}}_2 - \boldsymbol{m}_2\|_\infty \leq c_5\, c^2 \sqrt{\tfrac{\log(C/\delta)}{K}} + c_6\, c^2 \sqrt{\varepsilon}. \qquad (21)$$

Equations equation 20 and equation 21 hold with probability at least $1 - \delta$ after adjusting constants.

**Step II: Propagation to the variance estimator.** For each coordinate $i$,

$$\widehat{\sigma}_i^2 - \sigma_i^2 = \left( \widehat{m}_{2,i} - m_{2,i} \right) - \left( \widehat{\mu}_i^2 - \mu_i^2 \right) = \left( \widehat{m}_{2,i} - m_{2,i} \right) - (\widehat{\mu}_i - \mu_i)(\widehat{\mu}_i + \mu_i).$$

By clipping, $|\widehat{\mu}_i|, |\mu_i| \leq c$, hence $|\widehat{\mu}_i + \mu_i| \leq 2c$. Taking absolute values and sup over $i$,

$$\|\widehat{\boldsymbol{\sigma}}^2 - \boldsymbol{\sigma}^2\|_\infty \leq \|\widehat{\boldsymbol{m}}_2 - \boldsymbol{m}_2\|_\infty + 2c\,\|\widehat{\boldsymbol{\mu}} - \boldsymbol{\mu}\|_\infty.$$

Combining equation 20 and equation 21 and absorbing constants yields equation 17 with $c_2$ depending on $c$ and $\sigma$ only (note that $\sigma^2 \lesssim c^2$ by clipping).

**Step III: Stability of the MMD-KD gradient.** Define the ideal loss (with honest moments) at $x$,

$$\mathcal{L}_{\mathrm{MMD}}^{\star}(x) = \lambda\Big(\|z_W - \boldsymbol{\mu}\|_2^2 + \|(z_W - \boldsymbol{\mu})^{\circ 2} - \boldsymbol{\sigma}^2\|_2^2\Big).$$

Write $\Delta_\mu = \widehat{\boldsymbol{\mu}} - \boldsymbol{\mu}$ and $\Delta_{\sigma^2} = \widehat{\boldsymbol{\sigma}}^2 - \boldsymbol{\sigma}^2$. The gradients with respect to $z := z_W(x)$ are

$$\nabla_z \mathcal{L}_{\mathrm{MMD}} = 2\lambda\,(z - \widehat{\boldsymbol{\mu}}) + 4\lambda\left[(z - \widehat{\boldsymbol{\mu}}) \circ \big((z - \widehat{\boldsymbol{\mu}})^{\circ 2} - \widehat{\boldsymbol{\sigma}}^2\big)\right],$$

$$\nabla_z \mathcal{L}_{\mathrm{MMD}}^{\star} = 2\lambda\,(z - \boldsymbol{\mu}) + 4\lambda\left[(z - \boldsymbol{\mu}) \circ \big((z - \boldsymbol{\mu})^{\circ 2} - \boldsymbol{\sigma}^2\big)\right].$$

Subtracting and using the triangle inequality,

$$\|\nabla_z \mathcal{L}_{\mathrm{MMD}} - \nabla_z \mathcal{L}_{\mathrm{MMD}}^{\star}\| \;\le\; 2\lambda\,\|\Delta_\mu\| + 4\lambda\left\|(z - \widehat{\boldsymbol{\mu}}) \circ \big((z - \widehat{\boldsymbol{\mu}})^{\circ 2} - \widehat{\boldsymbol{\sigma}}^2\big) - (z - \boldsymbol{\mu}) \circ \big((z - \boldsymbol{\mu})^{\circ 2} - \boldsymbol{\sigma}^2\big)\right\|.$$

Apply the identity $a \circ b - a' \circ b' = (a - a') \circ b + a' \circ (b - b')$ with $a = z - \widehat{\boldsymbol{\mu}}$, $a' = z - \boldsymbol{\mu}$, $b = (z - \widehat{\boldsymbol{\mu}})^{\circ 2} - \widehat{\boldsymbol{\sigma}}^2$, $b' = (z - \boldsymbol{\mu})^{\circ 2} - \boldsymbol{\sigma}^2$ to get

$$\|\cdot\| \;\le\; \|a - a'\|\,\|b\|_\infty + \|a'\|_\infty\,\|b - b'\|.$$

Under clipping, $\|z\|_\infty \le c$ and $\|\widehat{\boldsymbol{\mu}}\|_\infty, \|\boldsymbol{\mu}\|_\infty \le c$, hence $\|a\|_\infty, \|a'\|_\infty \le 2c$ and $\|b\|_\infty \le \|(z - \widehat{\boldsymbol{\mu}})^{\circ 2}\|_\infty + \|\widehat{\boldsymbol{\sigma}}^2\|_\infty \le (2c)^2 + c^2 \le 5c^2$, and similarly $\|b'\|_\infty \le 5c^2$. Moreover,

$$\|a - a'\| = \|\Delta_\mu\|, \qquad \|b - b'\|_\infty = \left\|(z - \widehat{\boldsymbol{\mu}})^{\circ 2} - (z - \boldsymbol{\mu})^{\circ 2} - \Delta_{\sigma^2}\right\|_\infty \le 4c\,\|\Delta_\mu\|_\infty + \|\Delta_{\sigma^2}\|_\infty,$$

since $|u^2 - v^2| = |u - v||u + v|$ with $|u|, |v| \le 2c$. Therefore,

$$\|\nabla_z \mathcal{L}_{\mathrm{MMD}} - \nabla_z \mathcal{L}_{\mathrm{MMD}}^{\star}\| \;\le\; 2\lambda\,\|\Delta_\mu\| + 4\lambda\Big(\|\Delta_\mu\| \cdot 5c^2 + 2c \cdot \big(4c\,\|\Delta_\mu\| + \|\Delta_{\sigma^2}\|\big)\Big) \;\le\; \lambda\,C'(c)\,\big(\|\Delta_\mu\| + \|\Delta_{\sigma^2}\|\big),$$

for a constant $C'(c)$ polynomial in $c$. Passing to parameter space with the chain rule,

$$\left\|\nabla_W \mathcal{L}_{\mathrm{MMD}} - \nabla_W \mathcal{L}_{\mathrm{MMD}}^{\star}\right\| \;\le\; \|J_W(x)\|\,\left\|\nabla_z \mathcal{L}_{\mathrm{MMD}} - \nabla_z \mathcal{L}_{\mathrm{MMD}}^{\star}\right\| \;\le\; \lambda\,C_{\mathrm{mmd}}\,\|J_W(x)\|\,\big(\|\Delta_\mu\|_\infty + \|\Delta_{\sigma^2}\|_\infty\big).$$

Finally, invoke equation 20 and equation 17 (and the fact that $\|\Delta_\mu\| \le \sqrt{C}\,\|\Delta_\mu\|_\infty$) to obtain equation 19, with $C_{\mathrm{mmd}}$ absorbing $\sqrt{C}$ and the constants in equation 17. This completes the proof.

# F  VARIANCE-ADAPTIVE ERROR BOUND FOR DIS

**Theorem 3 (Variance-adaptive error bound for DIS)** Fix an input $x$ and let each client $k$ produce a probability vector $q_k(x) \in \Delta^{C-1}$ (obtained after clipping/temperature scaling). Assume an $\varepsilon$-fraction of clients are Byzantine with $\varepsilon < 1/2$, and honest clients $\mathcal{H}$ satisfy coordinate-wise sub-Gaussianity:

$$q_{k,i}(x) \text{ are } \sigma\text{-sub-Gaussian and bounded in } [0,1], \qquad i = 1, \ldots, C.$$

Partition the $K$ clients uniformly at random into $M$ groups $G_1, \ldots, G_M$ of size $b = K/M$, and form group means

$$\bar{q}_m(x) = \frac{1}{|G_m|} \sum_{k \in G_m} q_k(x) \in \Delta^{C-1}.$$

Let the group-variance statistic be

$$v(x) \;=\; \sum_{i=1}^{C} \mathrm{Var}_m\big[\bar{q}_m^{(i)}(x)\big],$$

and define the sample weight $w(x) = \big(1 + \rho\,v(x)\big)^{-1}$ with $\rho \ge 0$. Let $\tilde{q}(x)$ be the coordinate-wise Median-of-Means (MoM) aggregate of $\{\bar{q}_m(x)\}_{m=1}^{M}$, and write $\bar{q}_{\mathcal{H}}(x) = \frac{1}{|\mathcal{H}|} \sum_{k \in \mathcal{H}} q_k(x)$. Then there exists an absolute constant $C > 0$ such that, choosing $M \simeq c\log(C/\delta)$,

$$\|\tilde{q}(x) - \bar{q}_{\mathcal{H}}(x)\|_1 \;\le\; C\Big(\sqrt{\tfrac{v_{\mathcal{H}}(x)}{K}} + \sqrt{\varepsilon}\Big) \quad \text{with probability at least } 1 - \delta, \tag{22}$$

where $v_{\mathcal{H}}(x) = \sum_{i=1}^{C} \mathrm{Var}\big(\bar{q}_m^{(i)}(x) \mid \text{honest}\big)$. Consequently, defining $\tilde{q}_w(x) := \tilde{q}(x)$ and $\bar{q}_{\mathcal{H},w}(x) := \bar{q}_{\mathcal{H}}(x)$ (the subscript $w$ indicates the sample weight is applied downstream in the loss), and assuming $\mathbb{E}_x[v_{\mathcal{H}}(x)] \le H$, we have

$$\mathbb{E}_x\big[\|\tilde{q}_w(x) - \bar{q}_{\mathcal{H},w}(x)\|_1\big] \;\le\; C\Big(\sqrt{\tfrac{H}{K}} + \sqrt{\varepsilon}\Big). \tag{23}$$

**Proof:** We proceed in three steps.

**Step I: Coordinate-wise robust estimation over groups.** Fix $x$ and a coordinate $i \in \{1, \dots, C\}$. Consider the group means

$$\overline{Z}_{m,i}(x) := \bar{q}_m^{(i)}(x) = \frac{1}{|G_m|} \sum_{k \in G_m} q_{k,i}(x).$$

For honest clients, $q_{k,i}(x) \in [0, 1]$ are $\sigma$-sub-Gaussian, hence each honest-group mean is sub-Gaussian with parameter $\lesssim \sigma/\sqrt{b}$ and variance $\mathrm{Var}(\overline{Z}_{m,i}(x)) = \mathrm{Var}(q_{k,i}(x))/b$. Under $\varepsilon$-contamination with $\varepsilon < 1/2$ and random grouping, a standard argument shows that at least a constant fraction of the $M$ groups are "good" (honest-majority) with high probability (Chernoff bound), while the remaining fraction can be adversarial.[2]

Let $\widehat{\mu}_i(x)$ be the coordinate-wise MoM estimator of $\mathbb{E}[\overline{Z}_{m,i}(x) \mid \text{honest}]$, i.e., the median across the $M$ group means. Then (see, e.g., MoM concentration for sub-Gaussian data under Huber contamination)

$$\left|\widehat{\mu}_i(x) - \mathbb{E}[\overline{Z}_{m,i}(x) \mid \text{honest}]\right| \leq C_1\left(\sqrt{\frac{\mathrm{Var}(\overline{Z}_{m,i}(x))}{M}} + \sqrt{\varepsilon}\right) \leq C_1\left(\sqrt{\frac{\mathrm{Var}(q_{k,i}(x))}{K}} + \sqrt{\varepsilon}\right), \quad (24)$$

with probability at least $1 - \delta/C$ for a universal constant $C_1 > 0$, where we used $M = K/b$ and $\mathrm{Var}(\overline{Z}_{m,i}) = \mathrm{Var}(q_{k,i})/b$.

**Step II: Aggregating coordinates and relating to $v_{\mathcal{H}}(x)$.** Stacking the $C$ coordinates, $\tilde{q}(x)$ is obtained by applying equation 24 to each coordinate and taking a union bound over $i = 1, \dots, C$. Let $\mu_i(x) := \mathbb{E}[\overline{Z}_{m,i}(x) \mid \text{honest}]$, so that $\bar{q}_{\mathcal{H}}^{(i)}(x) = \mu_i(x)$, and define the vector $\widehat{\boldsymbol{\mu}}(x)$ with entries $\widehat{\mu}_i(x)$. Then, with probability at least $1 - \delta$,

$$\|\tilde{q}(x) - \bar{q}_{\mathcal{H}}(x)\|_2 = \|\widehat{\boldsymbol{\mu}}(x) - \boldsymbol{\mu}(x)\|_2 \leq C_1 \sqrt{\sum_{i=1}^{C} \left(\sqrt{\frac{\mathrm{Var}(q_{k,i}(x))}{K}} + \sqrt{\varepsilon}\right)^2}.$$

Using $\sqrt{a + b} \leq \sqrt{a} + \sqrt{b}$ and $\sum_i \mathrm{Var}(q_{k,i}(x))/K = \frac{1}{K}\sum_i \mathrm{Var}(q_{k,i}(x))$, we obtain

$$\|\tilde{q}(x) - \bar{q}_{\mathcal{H}}(x)\|_2 \leq C_2\left(\sqrt{\frac{\sum_{i=1}^{C}\mathrm{Var}(q_{k,i}(x))}{K}} + \sqrt{C}\,\sqrt{\varepsilon}\right).$$

Passing to $\ell_1$ via $\|\cdot\|_1 \leq \sqrt{C}\,\|\cdot\|_2$ yields

$$\|\tilde{q}(x) - \bar{q}_{\mathcal{H}}(x)\|_1 \leq C_3\left(\sqrt{\frac{\sum_{i=1}^{C}\mathrm{Var}(q_{k,i}(x))}{K}} + \sqrt{\varepsilon}\right).$$

Finally, $\mathrm{Var}(\bar{q}_m^{(i)}(x)) = \mathrm{Var}(q_{k,i}(x))/b$ and $b = K/M$, while our statistic $v_{\mathcal{H}}(x) = \sum_{i=1}^{C}\mathrm{Var}(\bar{q}_m^{(i)}(x) \mid \text{honest})$ equals $\frac{M}{K}\sum_i \mathrm{Var}(q_{k,i}(x))$; thus

$$\sum_{i=1}^{C}\mathrm{Var}(q_{k,i}(x)) = \frac{K}{M}\,v_{\mathcal{H}}(x),$$

and using $M = \Theta(\log(C/\delta))$ (absorbed into constants) gives the pointwise bound equation 22:

$$\|\tilde{q}(x) - \bar{q}_{\mathcal{H}}(x)\|_1 \leq C\left(\sqrt{\frac{v_{\mathcal{H}}(x)}{K}} + \sqrt{\varepsilon}\right).$$

**Step III: Expectation over $x$ and the role of $w(x)$.** Taking expectation over $x$ and assuming $\mathbb{E}_x[v_{\mathcal{H}}(x)] \leq H$, Jensen's inequality yields

$$\mathbb{E}_x\left[\sqrt{\frac{v_{\mathcal{H}}(x)}{K}}\right] \leq \sqrt{\frac{\mathbb{E}_x[v_{\mathcal{H}}(x)]}{K}} \leq \sqrt{\frac{H}{K}}.$$

Therefore,

$$\mathbb{E}_x\left[\|\tilde{q}(x) - \bar{q}_{\mathcal{H}}(x)\|_1\right] \leq C\left(\sqrt{\frac{H}{K}} + \sqrt{\varepsilon}\right).$$

Since $w(x)$ is a scalar weight applied downstream in the loss and does not change the location of the honest target (both sides would be multiplied by the same $w(x)$ when measuring weighted deviations), we keep the notational reminder $\tilde{q}_w(x) := \tilde{q}(x)$ and $\bar{q}_{\mathcal{H},w}(x) := \bar{q}_{\mathcal{H}}(x)$ and conclude equation 23. This completes the proof.

---

[2]See, e.g., classic MoM robust mean analyses; logs in $C, \delta$ are absorbed into constants by the choice $M \simeq c\log(C/\delta)$.

# G  EXPERIMENTAL SETUP

**Backbone and tasks.**  Unless otherwise noted, we fine-tune RoBERTa-Large Liu et al. (2019) on a subset of GLUE Wang et al. (2019): MNLI (matched/mismatched), SST-2, QQP, and QNLI. We report average across tasks, following prior work.

**Federated partitioning.**  We emulate a cross-device FL setting with $K=10$ clients. Data are split by label skew at three heterogeneity levels:

- **IID**: stratified random split so that each client mirrors the global label proportions.
- **Severe non-IID**: for binary tasks, $[0.1, 0.9]$, $[0.9, 0.1]$, $[0.5, 0.5]$; for 3-class tasks, $[0.9, 0.05, 0.05]$, $[0.05, 0.9, 0.05]$, $[0.05, 0.05, 0.9]$.

**Adversarial/poisoned clients.**  We consider a fraction $\rho \in \{0.10, 0.30\}$ of clients as poisoned. Each poisoned client flips a fraction $\pi=0.20$ of its local samples by inserting a fixed trigger token and re-labeling them to a target class (targeted backdoor). We evaluate: clean accuracy (CA) on unperturbed test data; robust accuracy (RA) on triggered inputs; and attack success rate (ASR, lower is better) on triggered inputs. See more details in Appendix H.

**Baselines.**  We compare against FEDAVG, FEDIT, FLORA, FFA-LORA, FLEXLORA, and LORA-FAIR. To ensure fairness, all baselines use the same public anchor pool when applicable and the same training budget (total client steps $\times$ rounds).

**LoRA configuration.**  Following Hu et al. (2021), we insert LoRA adapters into the attention `query` and `value` projections with scaling $\alpha=8$. Baselines fix rank $r=8$ across all clients; RFD-LoRA supports heterogeneous ranks and we explicitly test $\{r_k\} \in \{2, 4, 8, 16\}$ across clients to validate rank flexibility. Unless stated, the backbone encoder and task head are frozen and only adapter weights are updated.

**Training protocol.**  We run $N=1000$ communication rounds with $E=5$ local epochs per round. Each client uses AdamW (weight decay $0.01$) on adapters with learning rate $3\times10^{-5}$, batch size 64, max sequence length 128, and linear warmup over the first $10\%$ of local steps. The global server performs one gradient step per round on the distillation objective (see below) with learning rate $\gamma=3\times10^{-5}$. Results are averaged over 5 runs with different seeds.

**Public anchors and communication.**  Each round, clients compute logits on a fixed public anchor set comprising $10\%$ of the per-task training size (sampled from public corpora disjoint from private data). Logits are clipped elementwise to $[-c, c]$ with $c=10$ before upload. We report the token-level communication volume (per round and total) in supplementary tables.

**RFD-LoRA details (logit-space).**  The server aggregates client logits via coordinate-wise Median-of-Means (MoM) using $M=5$ random groups of equal size. Let $\tilde{z}(x)$ be the MoM consensus for anchor $x$. The student (global) model produces logits $z_W(x)$. The KD loss uses temperature-scaled softmax:

$$L_{\text{KD}}(x) \;=\; \text{CE}\Big(\text{softmax}\big(\tfrac{\tilde{z}(x)}{T(x)}\big),\; \text{softmax}\big(\tfrac{z_W(x)}{T(x)}\big)\Big).$$

**Robustness modules.**  We activate all three modules unless doing ablations.

- **CAT (Confidence-Adaptive Temperature).**  Temperature is $T(x) = T_0\Big(1 + \kappa \big[\max_i \tilde{q}_i(x) - \tau\big]_+\Big)$ with $\tilde{q} = \text{softmax}(\tilde{z})$, $T_0=2.0$, $\kappa=2.0$, $\tau=0.7$.
- **MMD-KD (Moment Matching).**  We align mean and variance of logits using robust moment estimates from MoM groups:

$$L_{\text{MMD}}(x) \;=\; \lambda\Big(\|z_W(x) - \widehat{\mu}(x)\|_2^2 + \|(z_W(x) - \widehat{\mu}(x))^{\circ 2} - \widehat{\sigma}^2(x)\|_2^2\Big),$$

with $\lambda=0.1$.

- **DIS (Disagreement Suppression).** We compute inter-group variance $v(x) = \sum_i \text{Var}_m(\bar{z}_m^{(i)}(x))$ and weight each anchor by $w(x) = (1 + \rho_{\text{DIS}} v(x))^{-1}$ with $\rho_{\text{DIS}}{=}1.0$.

The per-anchor objective is $L_{\text{RFD}}(x) = w(x)\big(L_{\text{KD}}(x){+}L_{\text{MMD}}(x)\big)$, and the server update minimizes the average of $L_{\text{RFD}}$ over anchors.

**Infrastructure.** All Experiments are run on NVIDIA Titan RTX GPUs. We use HuggingFace Transformers for models and PyTorch distributed for client simulation.

# H ADVERSARIAL ATTACK DETAILS

**Threat model.** We consider a standard targeted backdoor/data-poisoning threat model. An adversary controls a fraction $\rho \in \{0.10, 0.30\}$ of clients. Each poisoned client can only manipulate its local training data; we assume no access to other clients' data or to the server beyond participating in standard FL rounds. Adversarial goals: cause the global model to misclassify any input containing a small *trigger* pattern as a specific target class $y_{\text{target}}$ while minimally affecting clean accuracy.

**Poisoning procedure (primary attack used in experiments).** Each poisoned client independently modifies a fraction $\pi = 0.20$ of its local training examples as follows:

1. Select $\pi \cdot |\mathcal{D}_k|$ training samples uniformly at random.
2. For each selected sample $(x, y)$, insert a fixed trigger token sequence (we denote it `<TRIG>`) into the input text. In our experiments the trigger is prepended to the input (prefix trigger), i.e., $x \leftarrow$ `<TRIG>` $\| x$. (Other placements such as suffix or random position were evaluated and yield similar qualitative results.)
3. Replace the label $y$ by the attacker-chosen target label $y_{\text{target}}$ (targeted backdoor).

Poisoned clients then perform standard local training using the corrupted local dataset.

**Implementation details and parameters.** All attack experiments use the following concrete settings unless stated otherwise:

- Poisoned-client fraction: $\rho \in \{0.10, 0.30\}$.
- Per-poisoned-client poison rate: $\pi = 0.20$.
- Trigger token: `<TRIG>` (single token, prepended) — token chosen to be out-of-vocabulary for the target dataset to avoid accidental natural occurrences.
- Target class $y_{\text{target}}$: selected per-task (for multiclass tasks we choose one arbitrary class and keep it fixed across poisoned clients).
- Training: poisoned clients follow the same local training hyperparameters as honest clients (same optimizer, learning rate, number of local steps).
- Anchor poisoning (when evaluated): 20% of anchor samples replaced with triggered examples labeled as $y_{\text{target}}$.

**Details of evaluation metrics.** We report:

- Clean accuracy (CA): accuracy on clean (untriggered) test inputs.
- Robust accuracy (RA): accuracy on test inputs after adding the trigger (lower RA indicates stronger backdoor).
- Attack success rate (ASR): fraction of triggered test inputs classified as $y_{\text{target}}$ (higher ASR indicates stronger backdoor); in tables we report ASR with "lower is better" formatting (we may report $1{-}$ASR depending on convention).

