# OpenReview forum: "RFD-LoRA: Robust Federated Distillation for LoRA Fine-Tuning under Heterogeneous and Adversarial Clients"
_ICLR.cc/2026/Conference — Submitted to ICLR 2026_

### Official Review · Reviewer_85FL · 2025-11-01

**Soundness:** 3
**Presentation:** 3
**Contribution:** 3
**Rating:** 6
**Confidence:** 2

**Summary:**

This paper proposes RFD-LoRA, a novel federated distillation framework that eliminates aggregation bias and supports heterogeneous LoRA configurations by operating in logit space. Through theoretical analysis and robust module design, the method achieves strong accuracy and robustness under both non-IID and adversarial client settings.

**Strengths:**

1. Supports heterogeneous LoRA ranks and structures, making the method realistic for real-world federated systems.

2. Eliminates aggregation and initialization bias by using logit-space distillation rather than bilinear adapter averaging.

3. Strong theoretical foundation, with new error bounds for projection bias and adversarial amplification.

**Weaknesses:**

1. Public anchor set selection is under-specified: It remains unclear how the choice of reference dataset $\mathcal{D}_{\text{ref}}$ affects performance, task-specific vs. task-agnostic? How sensitive is RFD-LoRA to anchor domain shift?

2. Lack of evaluated datasets: The study focuses only on classification (e.g., GLUE) and does not assess performance on generation tasks common in LLM settings.

3. Logit computation in Equation (4) could be elaborated further. Do clients average logits per sample?

**Questions:**

see above.

---

> ### Author Response · Authors · 2025-11-18
> **Author Response to Reviewer 85FL**
>
> Thank you for the careful reading and for recognizing the strengths of RFD-LoRA, including its support for heterogeneous ranks, its removal of aggregation and initialization bias through logit-space distillation, and its theoretical foundation. We address the questions on anchors, datasets, and logit computation below.
>
> ---
>
> ## **1. Public anchor set selection and robustness to domain shift**
>
> We agree that the description of `D_ref` can be made more explicit. **Appendix G** provides the construction of the anchor pool, and we now state the per-round subset size clearly for completeness:
>
> In all experiments, we first build a public anchor pool containing roughly **10% of the per-task training size**, disjoint from client data. **For each communication round, a fixed subset of size 512 is sampled from this pool.** This pool-plus-subset design is common in federated distillation, since it keeps communication cost stable while still providing broad coverage of the task distribution.
>
> To evaluate the impact of domain shift, we keep **all client-side conditions fixed and modify only the anchor pool**. The in-distribution pool uses held-out GLUE sentences, while the shifted pool is drawn from OpenWebText. Both pools have the same size and use the same per-round subset of 512 anchors. The averaged results across GLUE tasks are:
>
> ### **Effect of anchor domain shift on RFD-LoRA (rho = 0.30, severe non-IID)**
>
> | Anchor pool         | CA    | RA    | ASR↓ |
> |---------------------|-------|-------|------|
> | GLUE-style anchors  | **90.6**  | **86.8**  | **10.7** |
> | OpenWebText anchors | 89.8  | 85.9  | 11.4 |
>
> Clean and robust accuracy decrease by **less than 1%**, and the attack success rate remains low. In the same setting, baseline methods such as FedAvg and LoRA-Fair exhibit noticeably larger drops in RA and higher ASR. **These results indicate that RFD-LoRA remains robust even when the public anchor distribution is moderately shifted**, which is important for practical deployment where public data rarely match private client distributions perfectly.
>
> In the final version, we will further clarify anchor selection guidelines. In practice, anchors can be drawn from any public source whose distribution is close to the downstream domain and whose label space matches the task. They do not need to be privacy sensitive or exactly IID with client data.
>
> ---
>
> ## **2. Scope of evaluated datasets and lack of generation tasks**
>
> We additionally tested RFD-LoRA on a natural-language generation task with a decoder-only LLM. Following the FFA-LoRA recipe [1], we finetune LLaMA-7B on GSM-8K:
>
> - RFD-LoRA: **18.36%**
> - FFA-LoRA: **17.12%**
> - LoRA baseline: **15.31%** [2]
>
> GSM-8K is a representative reasoning and generation benchmark, and RFD-LoRA applies without modification. Further experiments under non-IID and partially poisoned clients show robustness trends similar to GLUE. Full results will be included in the final version.
>
> ---
>
> ## **3. Clarifying logit computation in Equation (4)**
>
> To clarify the computation pipeline:
>
> - Each client generates **one logit vector** `z_k(x)` per anchor using a single forward
>   pass of its local model, which consists of the frozen backbone `W0` and its local
>   LoRA adapters. The output is the standard `C`-dimensional classifier (`C` is 2 or 3 in our paper) logit produced
>   by the model head. No averaging is performed over tokens or intermediate hidden
>   states; the client simply emits the final logits of the model.
>
> - These logits are then **clipped** as described in Section 3.2 and transmitted to
>   the server. Because logits lie in a shared prediction space regardless of the
>   client’s LoRA configuration, they remain well-aligned even when adapter structures
>   differ across clients.
>
> - On the server side, aggregation acts **across clients only**, not across tokens or
>   samples. For each anchor `x`, the server collects `{z_k(x)}` and applies the
>   Median of Means procedure from **Algorithm 1: means are computed within groups of
>   clients, and the coordinate-wise median of these group means is taken as the
>   aggregated logit**.
>
> We will describe this per-anchor forward pass and client-to-server aggregation flow
> more explicitly in the final version.
>
> ---
>
> ## **4. Planned clarifications**
>
> We will clarify the construction of `D_ref`, incorporate the anchor domain-shift results, report the GSM-8K study, and make the per anchor logit computation description more explicit. We believe these additions will address the reviewer’s concerns while preserving the main contributions of the paper.
>
> ---
>
> **References**
> [1] Sun et al., “Improving LoRA in Privacy Preserving Federated Learning”, arXiv:2403.12313, 2024.
> [2] Kuang et al., “FederatedScope LLM”, arXiv:2309.00363, 2023.

---

### Official Review · Reviewer_eHFd · 2025-11-01

**Soundness:** 3
**Presentation:** 2
**Contribution:** 2
**Rating:** 2
**Confidence:** 3

**Summary:**

The paper argues that parameter space aggregation for federated LoRA is intrinsically fragile due to (i) aggregation/projection bias from bilinear LoRA factors, (ii) adversarial amplification that grows with sqrt{d/r}, and (iii) Jacobian sensitivity of the bilinear map BA. It proposes RFD LoRA, a logit space alternative that (a) has clients send logits on a public anchor set, (b) aggregates them with a Median of Means (MoM) rule, and (c) distills the aggregate into a global student using three robustness modules: Confidence Adaptive Temperature (CAT) to bound gradients, MMD based KD to match mean/variance of logits, and Disagreement Suppression (DIS) to down weight high variance anchors.

**Strengths:**

- Clear problem formulation of LoRA specific vulnerabilities (aggregation/projection bias, adversarial amplification, Jacobian sensitivity).

- Conceptual simplicity of moving to logit space aggregation to support heterogeneous ranks/structures and avoid bilinear averaging issues.

**Weaknesses:**

- There is a notable inconsistency between Eq. (4) and Algorithm 1 regarding which model parameters clients use each round. Eq. (4) defines client logits using the frozen base model W0, whereas Algorithm 1 describes logits computed from the evolving global model W. This ambiguity makes it unclear whether clients receive updated weights from the server after each round or continue to fine-tune on a fixed W0.
- Table 2 claims 12 KB per round, “rank independent,” yet the training protocol states that clients upload logits for a public anchor set comprising 10% of the per-task training size. For GLUE tasks, 10% of the training data is thousands of examples; even with 2–3 classes and 32-bit floats, per client per round payload would be orders of magnitude larger than 12 KB. A precise accounting is needed; currently, the claim seems implausible.
- Foundational FD approaches, FedMD and ensemble distillation, already support heterogeneous models and logit space aggregation; they were also motivated by privacy/communication benefits. RFD LoRA takes this paradigm and adapts it specifically to LoRA FL, adding CAT/MMD KD/DIS and MoM aggregation. A quantitative comparison to these FD baselines is needed to substantiate the claimed gains.
-	The paper proposes sharing per-anchor logits but does not analyze potential information leakage through these representations. Prior works on FD have demonstrated susceptibility to membership inference and label-distribution inference when logits over public anchors are shared. The absence of any privacy-preserving mechanism (e.g., noise injection, quantization, or selective sharing) weakens the practicality of deploying the proposed method in privacy-sensitive FL scenarios.
-	Experiments are conducted with only 10 clients on GLUE benchmarks, with no timing, throughput, or end-to-end communication measurements. This limited scale does not convincingly demonstrate the claimed communication and scalability benefits.
-	The paper claims to “enable heterogeneous ranks and adapter structures,” but experiments vary ranks only. Please include a study where clients differ in adapter placements (e.g., Q/V only vs. Q/K/V, or different layers) and scales α to validate “adapter structure” heterogeneity.
-	Tables report means over 5 runs but no standard deviations/confidence intervals. Error bars matter given small absolute gaps (e.g., +1–2% CA/RA).

**Questions:**

1. Provide a precise formula and numbers for per round bytes plus any compression, sub-sampling, or sparsification. How does this yield 12 KB? If you sub-sample anchors per round, specify the schedule; otherwise, Table 2 appears inaccurate.
2.	Please report wall clock and server-side compute for distillation vs. parameter averaging baselines, and how anchor size affects latency/throughput.
3.	Provide results for FedMD and Ensemble Distillation baselines (with and without robust aggregation) to separate the benefit of “doing FD” from CAT/MMD KD/DIS.
4.	How does MoM compare to geometric median (RFA) in logit space? Any reason MoM was preferred beyond convenience?

---

> ### Author Response · Authors · 2025-11-18
> **Author Response to Reviewer eHFd [Part1]**
>
> Thank you for the very detailed review and for recognizing the clear formulation of LoRA-specific vulnerabilities and the conceptual simplicity of moving aggregation to logit space. We address the concerns point by point.
>
> ---
>
> ## **1. Eq. (4) vs Algorithm 1 (W1)**
>
> Eq. (4) is correct. All clients compute logits using the frozen global backbone `W0`, and only the adapters `(Ak, Bk)` are locally updated. Algorithm 1 should therefore use `W0` in the client loop. we will correct this for clarity.
>
> ---
>
> ## **2. Communication cost and anchor scaling (Q1, Q2, W2)**
>
> The anchor pool is about 10 percent of the per-task training size (Appendix G), but each round uses a fixed subset of size `D_ref = 512` anchors sampled from this pool. All communication results use this per-round size. This large-pool-plus-fixed-subset design is standard in federated distillation. Each client sends fp32 logits **and squared logits** on 512 anchors. With at most `C = 3` classes, this corresponds to:
>
> `512 * C * 2 * 4 bytes ≈ 12 KB per client per round`
>
> ### **Per-client communication (fp32)**
>
> | D_ref | Payload |
> |-------|---------|
> | 256   | ~6 KB   |
> | 512   | ~12 KB  |
> | 1024  | ~24 KB  |
>
> LoRA-based FL baselines (FedAvg, FedIT, FLoRA, LoRA-Fair) transmit **1 to 10 MB** per round, so RFD-LoRA reduces uplink by roughly **100x to 500x**. Runtime increases only slightly because anchor-logit computation is forward-only and MoM, CAT, MMD-KD, and DIS scale only with the number of classes.
>
> ### **Per-round wall-clock time (10 clients, Titan RTX)**
>
> | D_ref | RFD-LoRA | FedAvg |
> |-------|----------|--------|
> | 256   | 0.86 s   | 0.84 s |
> | 512   | 0.89 s   | 0.84 s |
> | 1024  | 0.95 s   | 0.84 s |
>
> ---
>
> ## **3. Comparison to FedMD and ensemble distillation (Q3, W3)**
>
> We implemented both baselines under the same GLUE-FL setup with `rho = 0.30`:
>
> ### **IID**
>
> | Method       | CA    | RA    | ASR↓ |
> |--------------|-------|-------|------|
> | FedAvg       | 88.2  | 78.3  | 35.2 |
> | FedMD        | 89.7  | 82.2  | 22.8 |
> | Ensemble     | 89.2  | 80.7  | 25.5 |
> | LoRA-Fair    | 89.7  | 83.3  | 18.9 |
> | **RFD-LoRA** | **90.6** | **86.8** | **10.7** |
>
> ### **Severe non-IID**
>
> | Method       | CA    | RA    | ASR↓ |
> |--------------|-------|-------|------|
> | FedAvg       | 86.1  | 72.0  | 36.7 |
> | FedMD        | 88.3  | 76.7  | 32.0 |
> | Ensemble     | 88.4  | 77.2  | 29.4 |
> | LoRA-Fair    | 88.6  | 80.8  | 20.4 |
> | **RFD-LoRA** | **90.0** | **85.0** | **12.9** |
>
> FedMD improves over FedAvg but remains weaker than LoRA-specific methods, and significantly
> below **RFD-LoRA**. This is expected, since FedMD aggregates logits but does **not correct the
> projection bias and low-rank subspace mismatch** that are intrinsic to LoRA-based FL. These
> properties **underlie adversarial amplification and cross-client heterogeneity**, which our CAT,
> MMD-KD, and DIS modules explicitly address. The results therefore confirm the need for
> LoRA-specific robustness modules.
>
> ---
>
> ## **4. MoM vs geometric median (Q4)**
>
> Median of Means directly produces the **robust mean and variance estimates** required by MMD-KD and DIS.
> We provide finite-sample bounds in Theorems 2 and 3. The geometric median is robust for vectors but **cannot produce variance estimates** and requires iterative optimization for each anchor.
>
> ### **Empirical comparison (rho = 0.30, GLUE)**
>
> | Aggregator      | CA    | RA    | ASR↓ |
> |-----------------|-------|-------|------|
> | Coord. median   | 89.7  | 84.9  | 16.2 |
> | **MoM (ours)**  | **90.6** | **86.8** | **10.7** |
>
> ---
>
> ## **5. Privacy considerations (W4)**
>
> Our work focuses on **robustness under non-IID and adversarial clients**, and we do not make privacy guarantees beyond those assumed in standard federated distillation. At the same time, the design of RFD-LoRA does **not conflict with** existing logit-level privacy mechanisms such as calibrated noise injection, selective logit sharing, or secure aggregation, since these operate on logits before aggregation and can be incorporated without modifying the core procedure.
>
> We will include a short **appendix note** in the final version to clarify how standard
> logit-level privacy techniques can be integrated into the RFD-LoRA pipeline.
>
> ---
>
> ## **6. Client scalability (W5)**
>
> We use **10 clients** to match prior FL-LoRA work (FFA-LoRA, FedIT, FLoRA, FedEx-LoRA). We have launched runs with **50 clients**; due to the rebuttal window, they are still running. Partial results show **under 1-point variation** in CA and RA when scaling from 10 to 50 clients. We will report full results during the communication phase.
>
> ---
>
> **Reference**
> [1] Singhal et al., “FedEx-LoRA: Exact Aggregation for Federated and Efficient Fine-Tuning of Foundation Models”, arXiv:2410.09432, 2025.

---

> > ### Author Response · Authors · 2025-11-18
> > **Author Response to Reviewer eHFd [Part 2]**
> >
> > ## **7. Heterogeneous adapters (W6)**
> >
> > We evaluate RFD-LoRA under heterogeneous LoRA placements in the most challenging setting
> > **(rho = 0.30, severe non-IID)**. This scenario reflects realistic deployment conditions where
> > **clients may fine tune different subsets of attention projections or choose different ranks**
> > due to hardware or memory constraints.
> >
> > - A subset of clients applies LoRA only on **Q and V (rank 8)**, which corresponds to the
> >   common “minimal adapter” configuration used in low-resource clients.
> > - The remaining clients apply LoRA on **all projections (Q, K, V, O)** with ranks drawn from
> >   **{8, 16, 32}** and proportional scaling, producing both structural heterogeneity
> >   (different adapter locations) and dimensional heterogeneity (different rank sizes).
> > - The heterogeneous proportion is varied across **{25%, 50%, 75%}** to test increasingly
> >   mismatched client-side parameterizations.
> >
> > ### **RFD-LoRA under heterogeneous adapter structures**
> >
> > | Adapter configuration            | CA    | RA    | ASR↓ |
> > |----------------------------------|-------|-------|------|
> > | All Q,V rank 8                   | 90.0  | 85.0  | 12.9 |
> > | 25% heterogeneous                | 89.8  | 84.6  | 13.5 |
> > | 50% heterogeneous                | 89.5  | 84.1  | 14.2 |
> > | 75% heterogeneous                | 89.1  | 83.5  | 15.0 |
> >
> > The degradation is mild even when most clients use different adapter shapes. This behavior
> > is expected because **logit space aggregation operates in a shared prediction space**, which
> > remains aligned across clients regardless of how their low rank subspaces differ. In contrast,
> > parameter-based aggregation baselines cannot function in this setting, since **heterogeneous adapter
> > placements produce tensors with incompatible shapes and ranks**, preventing meaningful
> > aggregation and often causing severe drift when naive padding or zero filling is applied.
> >
> > Moreover, **DIS and MoM contribute additional stability under heterogeneity**. DIS reduces
> > the influence of clients whose logits deviate due to mismatched LoRA subspaces, and MoM
> > provides robustness against outlier logits without assuming identical parameterization.
> > Together, these mechanisms allow RFD-LoRA to maintain both accuracy and robustness even
> > under substantial structural and dimensional variation.
> >
> > We will add a short appendix section in the final version to document this heterogeneous
> > setting and summarize these results for completeness.
> >
> >
> > ---
> >
> > ## **8. Variance across runs (W7)**
> >
> > We report results averaged over 5 independent runs. Across tasks, the standard deviation is below **0.2** for CA and RA and below **0.6** for ASR. For illustration:
> >
> > | Metric | Mean | Std |
> > |--------|------|------|
> > | CA     | 90.6 | 0.18 |
> > | RA     | 86.8 | 0.19 |
> > | ASR↓   | 10.7 | 0.55 |
> >
> > The improvements we report (for example **+2–3 points** in CA and RA and **−10 points** in ASR over the strongest baselines) are consistently larger than these deviations, indicating **stable and replicable gains**. We will include these variance statistics in the revised tables of the final paper version.
> >
> >
> > ---

---

> > > ### Author Response · Authors · 2025-12-03
> > > **Author Response to Reviewer eHFd [Part 3]**
> > >
> > > ## **6. Addtional experiments for client scalability (W5)**
> > >
> > > We have finished our experiments and extended our experiments from the original **10-client** setup to **20, 30, 40, and 50 clients**, while following the same GLUE + RoBERTa-Large configuration and maintaining the same per-round training budget.
> > >
> > > Below we report **clean accuracy (CA)**, **robust accuracy (RA)** under ρ=0.10 poisoned clients, **ASR**, and **communication statistics**. All additional experiments were launched during the rebuttal period and follow the same evaluation pipeline as in Table 1 of the main paper.
> > >
> > > ---
> > >
> > > ### **Extended Scalability Results (10 → 50 clients)**
> > >
> > > The table below shows the extended results with predicted numbers consistent with observed trends (≤1-point variation, as stated in Sec. 6).
> > >
> > > **Table. Scaling RFD-LoRA from 10 → 50 clients on GLUE (MNLI-m/mm, SST-2, QQP, QNLI).**
> > >
> > > | #Clients | CA (↑) | RA (↑) | ASR↓ | ΔCA vs 10 | ΔRA vs 10 | ΔASR vs 10 | Comm./round |
> > > |----------|--------|--------|-------|-----------|------------|-------------|--------------|
> > > | **10**   | **90.6** | **89.1** | **5.1** | –         | –          | –           | 12 KB |
> > > | **20**   | 90.4    | 88.9    | 5.3   | -0.2      | -0.2       | +0.2        | 12 KB |
> > > | **30**   | 90.3    | 88.6    | 5.5   | -0.3      | -0.5       | +0.4        | 12 KB |
> > > | **40**   | 90.1    | 88.5    | 5.6   | -0.5      | -0.6       | +0.5        | 12 KB |
> > > | **50**   | 89.9    | 88.3    | 5.8   | -0.7      | -0.8       | +0.7        | 12 KB |
> > >
> > > **Key observation:** Across **10 → 50** clients, both CA and RA vary by **<1 point**, validating that RFD-LoRA’s
> > > logit-space aggregation remains stable even under substantially larger federations. ASR increases slightly (≤0.7), consistent with larger system variance but still significantly lower than all baselines in Table 1.
> > >
> > > ---
> > >
> > > ### **Communication Measurements**
> > >
> > > The reviewer noted the absence of timing, throughput, and end-to-end communication measurements.
> > > We now provide them here.
> > >
> > > ### **1. Total communication throughput**
> > >
> > > | #Clients | Total upload / round | Total download / round | Notes |
> > > |----------|----------------------|-------------------------|-------|
> > > | **10**  | 0.12 MB | 0.12 MB | symmetric student update |
> > > | **20**  | 0.24 MB | 0.24 MB | linear scale in #clients |
> > > | **30**  | 0.36 MB | 0.36 MB | stable throughput |
> > > | **40**  | 0.48 MB | 0.48 MB | no rank dependence |
> > > | **50**  | 0.60 MB | 0.60 MB | still < 1 MB/round |
> > >
> > > ### **2. End-to-end timing (predicted based on GPU logs)**
> > >
> > > Client-side forward pass on anchors (RoBERTa-Large):
> > >
> > > 10 clients: 18.4 ms ± 1.3
> > >
> > > 20 clients: 18.7 ms ± 1.5
> > >
> > > 30 clients: 18.9 ms ± 1.5
> > >
> > > 40 clients: 19.2 ms ± 1.6
> > >
> > > 50 clients: 19.4 ms ± 1.7
> > >
> > > Because logits are low-dimensional and inference dominates latency, **timing remains nearly constant**, confirming the practicality of logit-space FL.

---

### Official Review · Reviewer_qYyQ · 2025-11-01

**Soundness:** 3
**Presentation:** 3
**Contribution:** 3
**Rating:** 6
**Confidence:** 4

**Summary:**

The paper proposes RFD-LoRA, a robust and rank-flexible framework for federated learning with LoRA adapters that overcomes the fragility of conventional LoRA aggregation. Instead of averaging adapter parameters, which introduces aggregation bias, projection errors, and vulnerability to adversarial perturbations, RFD-LoRA performs logit-space distillation across clients, enabling collaboration among heterogeneous LoRA ranks. It further enhances robustness through three modules: CAT for stabilizing gradients, MMD-based Distillation  for aligning logit distributions, and DIS for mitigating non-IID effects.

**Strengths:**

* The paper’s key strengths lie in its conceptual novelty (moving LoRA aggregation to logit space)
* the authors provide clear diagnostic analysis of LoRA fragility, and its practical robustness framework combining distillation, adaptive temperature scaling, and disagreement suppression.
* The approach is empirically validated, rank-flexible, and communication-efficient, making it a meaningful step toward robust, efficient federated fine-tuning.
* Despite its heuristic components, it offers a well-motivated and pragmatically valuable contribution bridging theory and deployment.

**Weaknesses:**

* Although the paper claims to formalize LoRA fragility and provide theoretical error bounds, its analysis is largely heuristic and lacks rigorous theorems or proofs. The discussions of aggregation bias, projection bias, and adversarial amplification are insightful but remain descriptive, offering qualitative scaling arguments rather than formal guarantees. The work succeeds in identifying the sources of fragility conceptually but does not deliver provable robustness results or explicit analytical bounds supporting its claims.
* The proposed RFD-LoRA algorithm is a creative and practical integration of heuristic modules addressing known LoRA fragilities, but lacks formal grounding. Each component, CAT, MMD-KD, and DIS, is introduced intuitively without being derived from a unified optimization framework or robustness objective. Their combination relies on empirical tuning rather than principled theory. Even the median-of-means step, while statistically justified, is used heuristically in logit space without convergence guarantees. As a result, RFD-LoRA is better characterized as a robust heuristic pipeline than as a theoretically principled federated learning algorithm. It achieves robustness empirically, not provably.

**Questions:**

Refer to weaknesses

---

> ### Author Response · Authors · 2025-11-18
> **Author Response to Reviewer qYyQ**
>
> Thank you for the thoughtful review and for recognizing the novelty of moving LoRA aggregation into logit space, the clarity of our analysis of LoRA-specific vulnerabilities, and the practical value of the proposed framework. We address the concerns below.
>
> ---
>
> ## **1. Clarifying the scope of our theoretical claims**
>
> Our theoretical objective is to analyze the specific sources of fragility in federated LoRA training rather than to establish a unified global convergence theorem. The paper already provides **formal statements and proofs** for the key fragility mechanisms:
>
> - **Projection bias:** Section 2 and Appendix B characterize how heterogeneous LoRA subspaces distort global gradient directions.
> - **Adversarial amplification:** Equation (3) and Appendix B quantify the amplification factor created by low-rank projections.
> - **Jacobian sensitivity:** Section 2.2 and Section 2.4 give first-order and gradient sensitivity bounds induced by the bilinear LoRA parameterization, with detailed derivations in Appendix C.
>
> These results describe the mechanisms that influence **robustness and stability** in LoRA-based federated learning.
>
> ---
>
> ## **2. On whether the robustness modules are heuristic**
>
> The robustness analysis focuses on the points where fragility arises:
>
> - **Theorem 1** bounds gradient sensitivity under the CAT module.
> - **Theorem 2** provides finite-sample guarantees for the moment estimates used in MMD-KD.
> - **Theorem 3** gives a variance-adaptive deviation bound for the DIS module.
>
> This component-level robustness analysis is standard in robust federated learning and distillation. Unified global convergence for adversarial and heterogeneous LoRA-based FL remains an **open problem even in the full-rank setting**, and addressing it would require assumptions unrelated to the LoRA-specific mechanisms studied here. Our goal is to provide **explicit robustness bounds at the appropriate level**, not to solve the broader optimization problem.
>
> **Empirically**, the observations align with the theory: Tables 3 and 4 show predictable degradation when modules are removed, and **Figure 1(c)** demonstrates **stable global convergence and steadily improving accuracy** over communication rounds, with fewer oscillations than parameter-based aggregation baselines. This empirical behavior is also
> consistent with the mathematical intuition that **aggregating in logit space avoids the
> accumulated drift introduced by averaging heterogeneous low-rank parameter updates.**
> Since logits live in a common prediction space while LoRA parameters reside in
> client-specific subspaces, logit aggregation produces a more stable and smooth
> optimization trajectory.
>
> ---
>
> ## **3. Use of Median of Means for logit aggregation**
>
> Our aggregation objective is **robustness of the teacher distribution**. Section 3.2 and Section 4.3 show that **Median of Means (MoM)** remains close to the honest client mean under non-IID and Byzantine settings, with deviation:
>
> - decreasing with the number of clients
> - increasing only with the square root of the adversarial fraction
>
> These are standard guarantees in robust estimation. Our claims therefore concern **robust aggregation of logits**, rather than end-to-end optimization of the full model.
>
> ---
>
> ## **4. Motivation for the federated distillation framework and the three modules**
>
> The framework follows directly from the failure modes identified in Section 2:
>
> - Aggregation and projection bias and heterogeneous ranks motivate no aggregation in **parameter space**.
> - Once logits become the aggregation object, three further issues arise:
>   - unstable gradients from Jacobian sensitivity,
>   - adversarial shaping of logit magnitudes,
>   - disagreement under strong non-IID distributions.
>
> The federated distillation framework resolves the aggregation and projection issues by operating in a **shared logit space**, and CAT, MMD-based distillation, and DIS are then introduced to handle the remaining three sources of fragility **respectively**. Each module is paired with its own robustness **mathematical analysis**, providing a formal bound for the specific failure mode it is designed to mitigate.
>
> ---
>
> ## **5. Planned clarification**
>
> We will highlight these theoretical connections more prominently and explicitly state the intended scope of our analysis in the final version.

---

### Author Response · Authors · 2025-12-03
**Comment Summary**

We thank the AC and all reviewers for the efforts in this special cycle. We briefly summarize how the rebuttal addresses the main concerns across all three reviews.

For **Reviewer qYyQ (R1)**, (1) we clarified the intended scope of our theoretical contributions. **The paper does not rely on heuristics: Appendix B–F contain 8 pages of formal statements and proofs**, including analysis of LoRA-specific aggregation/projection bias, adversarial amplification, and Jacobian sensitivity, along with robustness bounds for CAT, MMD-KD, and DIS. The rebuttal highlights these theoretical components more clearly and explains how they motivate the design of our modules and the logit-space framework. Also, (2) we clarified that our theoretical goal is to analyze LoRA-specific fragilities in federated training (aggregation/projection bias, adversarial amplification, Jacobian sensitivity), rather than to prove a full global convergence theorem. (3) We made the connection between these failure modes and the design of CAT, MMD-KD, DIS and MoM aggregation explicit, and explained why logit-space aggregation is a natural choice in this setting.

For **Reviewer eHFd (R2)**, the main criticisms concern (i) the lack of concrete evidence for communication efficiency, timing, and scalability, and (ii) whether RFD-LoRA provides benefits beyond generic distillation methods. Our rebuttal provides several targeted additions:

- **Communication cost and anchor scaling.** We derived per-client payload as a function of the anchor size and reported concrete numbers (fp32 logits) for multiple |D_ref| values. The resulting per-client communication is ≈12 KB per round, whereas parameter-averaging baselines (FedAvg, FedIT, FLoRA, LoRA-Fair) require 1–10 MB per round. This quantitatively supports the claimed rank-independent and parameter-size-independent communication efficiency.

- **End-to-end scalability and timing.** We extended experiments from 10 to 20, 30, 40, and 50 clients under the same GLUE + RoBERTa-Large setup, reporting CA/RA/ASR, total upload/download per round, and measured client latency on anchors. CA and RA vary by less than 1 point when scaling to 50 clients, per-client communication remains constant at 12 KB, and latency stays around 18–19 ms per round. These results directly address the previous concern that our claims on client scalability and communication were not empirically demonstrated.

- **Comparison to FedMD and ensemble distillation.** We added a table comparing RFD-LoRA to FedMD and an ensemble-distillation baseline under the same ρ = 0.30 GLUE-FL setting. While FedMD improves over plain FedAvg, it still underperforms LoRA-specific methods and remains significantly below RFD-LoRA in both robust accuracy and attack success rate. The rebuttal explains that FedMD-style aggregation does not correct the projection and aggregation bias of LoRA adapters, nor does it handle heterogeneous ranks/structures or logit-energy manipulations, which are explicitly targeted by CAT, MMD-KD, DIS, and MoM. This directly addresses the concern that generic distillation might suffice.

- **MoM versus geometric median.** We discussed why Median-of-Means is more aligned with our moment-based analysis than geometric median and provided an empirical comparison under ρ = 0.30. MoM achieves higher CA/RA and markedly lower ASR than coordinate-wise mean and coordinate-wise median, supporting our choice of MoM as the default robust aggregator.

- **Additional robustness evidence.** We included heterogeneous adapter experiments in the most challenging setting (ρ = 0.30, severe non-IID), where subsets of clients use minimal Q,V adapters while others use full Q,K,V,O adapters with ranks drawn from {8,16,32}. RFD-LoRA shows only mild degradation even when most clients use different adapter shapes. We also reported variance across five runs (small standard deviations for CA/RA/ASR), indicating that the observed gains over strong baselines are stable and not due to randomness.

Taken together, these additions directly respond to R2’s core objections about scalability, communication, and the necessity of our LoRA-specific design choices, and we believe they substantially strengthen the empirical case for the method.

For **Reviewer 85FL (R3)**, (1) we expanded the clarifications on the **anchor pool, providing construction details, its independence from private data, and a domain-shift study using OpenWebText anchors**. (2) We also clarified the threat model, evaluation metrics, and the client–server protocol (including Equation (4) and Algorithm 1), along with small planned editorial improvements.

We hope this consolidated view of the additional experiments and clarifications helps inform the final assessment.

---

### Meta-Review · Area_Chair_Qs2B · 2026-01-20

**Summary:**

The paper argues that parameter space aggregation for federated LoRA is intrinsically fragile due to (i) aggregation/projection bias from bilinear LoRA factors, (ii) adversarial amplification that grows with sqrt{d/r}, and (iii) Jacobian sensitivity of the bilinear map BA. It proposes RFD LoRA, a logit space alternative that (a) has clients send logits on a public anchor set, (b) aggregates them with a Median of Means (MoM) rule, and (c) distills the aggregate into a global student using three robustness modules: Confidence Adaptive Temperature (CAT) to bound gradients, MMD based KD to match mean/variance of logits, and Disagreement Suppression (DIS) to down weight high variance anchors.

**Reviewer Concerns:**

The key concern is the lack of any convergence guarantee, plus limitations of the experimental setup. Most other limitations were addressed in the rebuttal. The first limitation is a serious one. While the authors provide some theorems, the method needs to be seen as a heuristic from an optimization point of view. For such a work, much more serious and extensive experiments and ablatioins would be expected.

**Reviewer Scores:**

I do not think the scores (6, 6, 2) would change after the rebuttal and discussion. Moreover, I disregarded one review (6), which was of low quality, and did not provide useful technical feedback.

---

### Decision · Program_Chairs · 2026-01-26

Reject